# Deconstructing the modular organization and real-time dynamics of mammalian spinal locomotor networks

Li-Ju Hsu[1,2], Maëlle Bertho [1,2] & Ole Kiehn [1,2] ✉

Locomotion empowers animals to move. Locomotor-initiating signals from the brain are funneled through descending neurons in the brainstem that act directly on spinal locomotor circuits. Little is known in mammals about which spinal circuits are targeted by the command and how this command is transformed into rhythmicity in the cord. Here we address these questions leveraging a mouse brainstem-spinal cord preparation from either sex that allows locating the locomotor command neurons with simultaneous $Ca^{2+}$ imaging of spinal neurons. We show that a restricted brainstem area – encompassing the lateral paragigantocellular nucleus (LPGi) and caudal ventrolateral reticular nucleus (CVL) – contains glutamatergic neurons which directly initiate locomotion. $Ca^{2+}$ imaging captures the direct LPGi/CVL locomotor initiating command in the spinal cord and visualizes spinal glutamatergic modules that execute the descending command and its transformation into rhythmic locomotor activity. Inhibitory spinal networks are recruited in a distinctly different pattern. Our study uncovers the principal logic of how spinal circuits implement the locomotor command using a distinct modular organization.

Locomotion—walking, flying, and swimming—is a motor action used in all animals to interact with the environment. The executing locomotor circuits that control coordinated muscle activity are localized in the spinal cord[1–3], whereas the commands for initiating locomotion reside in supraspinal structures, including the mesencephalic locomotor region (MLR), which control the start and speed of locomotion[4–11]. Reticulospinal command neurons integrate the signals from the MLR before they activate spinal executing locomotor circuits[5,11–14].

Early studies suggested that neurons in the medullary reticular formation (MRF) mediate the MLR locomotor command in cats and rats[12,13,15–19] and in the reticular formation in lamprey[20] and tadpole[21]. In mammals, some studies suggested that serotonergic neurons in MRF may mediate the descending command[5,22] while other studies pointed toward glutamatergic neurons being responsible for locomotor initiation[5,13,23]. There is also strong evidence that excitatory reticulospinal pathways are important for initiating vertebrate swimming[1,21].

The glutamatergic nature of the descending command neurons in mammals was confirmed directly by broad brainstem optogenetic stimulation[24] in newborn mice and was solidified by showing that cell- and target-specific activation of glutamatergic neurons in the lateral paragigantocellular nucleus (LPGi) induce fast locomotion in adult mice[25]. Anatomically glutamatergic LPGi neurons terminate in laminae VII–VIII of the mammalian spinal cord where the locomotor networks reside[3,26–29]. It is these spinal locomotor networks—composed of inhibitory and excitatory interneurons that generate the rhythm, left–right coordination, and flexor–extensor coordination in limbed animals— that must receive and implement the descending locomotor command to initiate locomotion. However, it remains unknown which neuronal populations receive the brainstem locomotor command and in which sequence the signal is transformed into rhythmic activity. Such information could reveal the functional composition of spinal locomotor networks. To address this question, it requires recording of system-wide activity pattern of identifiable neurons in the spinal cord with

[1]Department of Neuroscience, University of Copenhagen, 2200 Copenhagen, Denmark. [2]Department of Neuroscience, Karolinska Institutet, 171 77 Stockholm, Sweden. ✉e-mail: Ole.Kiehn@sund.ku.dk

spatial accuracy which is impossible to perform in vivo with any of the current technologies.

In this study, we, therefore, developed an in vitro brainstem-spinal cord preparation from the neonatal mouse to directly examine how descending locomotor initiating commands are executed by spinal networks. With cell-type-specific stimulation, we show that glutamatergic neurons in the LPGi and its neighboring caudal ventrolateral reticular nucleus (CVL) initiate locomotion via direct action on spinal locomotor circuits. Simultaneous stimulation of LPGi/CVL and large-scale $Ca^{2+}$ imaging from excitatory neurons in the spinal cord allowed us to reveal specific functional modules which exhibit a stereotyped spatial organization and a temporal activation sequence. By revealing the dynamics of excitatory functional modules, we show where the descending command arrives in the spinal cord and how this signal is transformed into rhythmic locomotor-like activity. Inhibitory networks are recruited in a sequentially different pattern. These data uncover the principal organization of how spinal circuits implement the locomotor command using a distinct modular organization and thus reveal a layered composition of spinal locomotor circuits.

## Results

### LPGi/CVL neurons initiate locomotor-like activity directly

To search the brainstem for an exact location of neurons that directly activate the spinal locomotor circuits, we developed an in vitro brainstem-spinal cord preparation of neonatal mice that allowed us to visually employ micro-electrical stimulation (mES) of defined areas in the brainstem while monitoring spinal motor output (Fig. 1a, c). Previous in vitro experiments have used broad stimulation of the brainstem to evoke locomotor-like activity (stimulation strength: 1–10 mA[30,31]). Here we minimized the stimulation intensity (20 μA; 9 Hz; 0.5 ms) to the range that was previously employed by reticulospinal brainstem mapping[32–34] with the aim of targeting relatively small areas of the brainstem. In accordance with these later studies, we found that the activated area (as measured by activity of excitatory neurons as shown in Supplementary Fig. 1) by our employed mES in the brainstem is restricted to an area with a diameter of $342 \pm 75$ μm ($N = 5$), smaller than that of stronger stimulation (Supplementary Fig. 1c, d).

We recorded ventral roots from the second (L2) and/or the fifth (L5) segments of the lumbar spinal cord to monitor the motor response evoked by the stimulation (Fig. 1c). In the mouse, drug-induced or hindbrain-evoked lumbar locomotor-like activity is characterized by alternating rhythmic bursting in ipsilateral flexor-dominant L2 and extensor-dominant L5 ventral roots and alternation between ipsi- and contra-lateral L2 (or L5) ventral roots, respectively[30,35,36]. These ventral root activities reflect coordinated locomotion observed in intact animals[37].

We used unilateral mES from the rostral to the caudal medulla to evoke locomotor-like activity (Fig. 1a, b and Supplementary Fig. 2). Unilateral mES was employed as it allowed us to systematically map the active brain-sites similar to previous brainstem mapping experiments[7,17,18,25,34,38]. The brainstem was consecutively sectioned rostrally to caudally in the transverse plane for easy access to map the stimulation sites dorsoventrally and mediolaterally (Fig. 1b, c). To accurately identify the rostrocaudal level of the section, we used crosses of $ChAT^{Cre};R26R^{tdTomato}$ neonatal mice, where cranial motor nuclei, e.g., the motor trigeminal nucleus (5 N), the facial nucleus (7 N), the dorsal motor nucleus of vagus (10 N), and the hypoglossal nucleus (12 N) (Fig. 1a, b), are distinctly visualized (Fig. 1d–g). A locomotor index was used to score the evoked ventral root activity (Supplementary Fig. 2b) and map the mES-effective site (Supplementary Fig. 2a). Among the four targeted rostrocaudal levels (Fig. 1d–g and Supplementary Fig. 2, $N = 17$), the level at the rostral 12 N (Fig. 1f) contained the most effective sites for initiating rhythmic activity compared to other levels (Fig. 1d, e, g and Supplementary Fig. 2c). At this level, rhythmic ventral root activity was consistently evoked by

unilateral mES at three clusters (the warm-colored areas in Fig. 1f) and not from nearby (350–400 μm) locations. The first cluster was located ventrally in the CVL with a weaker neighboring nucleus, the LPGi. The second cluster was situated laterally within the spinal trigeminal nuclei (Sp5). When stimulated these two clusters evoked alternation between left and right L2 roots activity, as well as L2 and L5 roots (flexor and extensor) activity on the same side of the spinal cord (Fig. 1m, n). The third cluster was located in the dorsal area of the gigantocellular reticular nucleus (Gi), just ventral to 12 N. This cluster, although effective in evoking rhythm activity, did not produce alternating locomotor activity (Supplementary Fig. 2c).

To evaluate whether these three clusters contained neurons that send a final locomotor command to the spinal cord, we carried out split-bath experiments that allow us to block collateral glutamatergic synaptic activity in the brainstem and the cervical and upper thoracic cord (Fig. 1h). The brainstem was perfused separately from the lumbar spinal cord by building a hydrophobic barrier at thoracic level 8 (Th8). We blocked glutamatergic neurotransmission in the brainstem and rostral spinal cord with 4 mM kynurenic acid (KA), which efficiently blocks NMDA and AMPA/kainate receptors[24,39,40].

In the presence of KA, the level at rostral 12 N (Fig. 1k, $N = 4$) remained more efficient in evoking rhythmic activity than other levels (Fig. 1i, j, l, $N = 13$, Supplementary Fig. 3a). mES of the first cluster corresponding to LPGi/CVL consistently evoked alternating locomotor-like patterns (Fig. 1k and Supplementary Fig. 3a). mES of the second cluster in Sp5, on the other hand, was completely ineffective in evoking locomotor bursts after the KA application (Fig. 1k and Supplementary Fig. 3a), suggesting that stimulation of this site—with intact glutamatergic transmission—activated the spinal locomotor circuits indirectly via other brainstem descending neurons. mES of the third cluster located in the dorsal Gi was still able to evoke rhythmic activity in the presence of KA, but without left-right alternating bursts (Fig. 1k and Supplementary Fig. 3a). Moreover, the mES-activated area in the brainstem is restricted to the LPGi/CVL region (Supplementary Fig. 1). These data indicate that the dominant population of neurons that initiate locomotion reside in LPGi and CVL (LPGi/CVL), which corresponds to the area found in adult mice[25].

The continuous unilateral stimulation of LPGi/CVL evoked rhythmic bouts of activity with alternation between left and right (lL2/rL2) and between rL2 (flexor) and rL5 (extensor) roots with coordination similar to that observed during drug application (Fig. 1m). The locomotor activity was sustained throughout the stimulation and lasted for a few cycles after the stimulation was terminated. The burst frequency during stimulation was high ($1.27 \pm 0.09$ Hz, range: 0.79–2.02 Hz, $N = 6$, stimulation frequency 5–20 Hz), compared with previously reported frequencies of drug-induced locomotor-like activity in mice (0.2–0.5 Hz)[29,35,36], but similar to what has been observed with broad optogenetic stimulation of glutamatergic brainstem neurons ($1.27 \pm 0.06$ Hz)[24]. The patterning between left–right and flexor–extensor activity is finely tuned to alternation, as demonstrated by the time-frequency (Fig. 1n, left panel) and the circular plot analyses (Fig. 1n, right panel, $N = 6$). Moreover, increasing stimulation frequency promoted locomotor activity at higher frequencies (Fig. 1o, one-way ANOVA, $N = 5$, $p < 0.05$).

These results show that LPGi/CVL at the level of rostral 12 N contains neurons that send a final descending command signal to evoke locomotor-like activity in the lumbar spinal cord.

### Locomotor-initiating LPGi/CVL neurons are glutamatergic

Since both glutamatergic and serotonergic brainstem neurons have been implicated in contributing to initiating locomotion[22,25], we examined if these two types of neurons were located in the LPGi/CVL at the level of rostral 12 N in neonatal mice. First, we performed RNA-scope® in situ hybridization to target the vesicular glutamate transporter type 2 (*Vglut2*) to label glutamatergic neurons and the

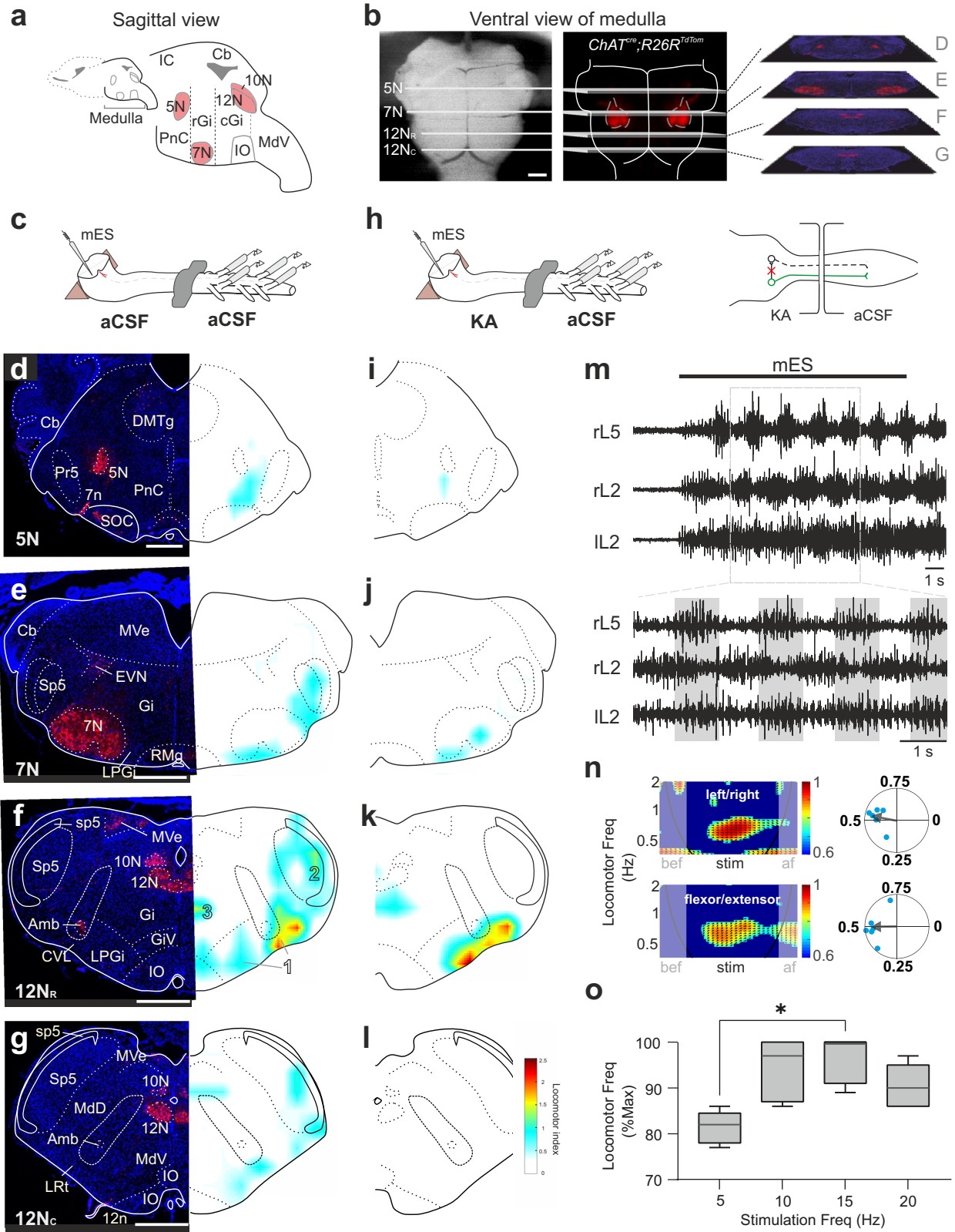

serotonergic transporter (*Sert*) to label serotonergic neurons (Fig. 2a, b). In accordance with a previous study[25], glutamatergic neurons were distributed in most of the ventral regions of the reticular formation, the LPGi, and the ventral part of the gigantocellular nucleus (GiV), Gi, as well as in CVL (Fig. 2a, b, *n* = 10 sections, *N* = 2 animals). Serotonergic neurons were located in the medial part of the LPGi and in the raphe nuclei (Fig. 2a, b). These results show that the

glutamatergic neurons are distributed within the area of LPGi/CVL, while the serotonergic neurons are located more medially.

Next, we performed anatomical tracing in *Vglut2^cre^;R26R^YFP^* neonatal mice with bilateral injections of the retrograde marker Cholera Toxin B (CTB) in the lower thoracic and upper lumbar segments of the spinal cord (T12–L2; Fig. 2c), where the critical elements of executing spinal locomotor networks are located[37,41]. CTB+ neurons were present

**Fig. 1 | Neurons located in LPGi/CVL send a final locomotor command to spinal circuits. a, b** Sagittal (**a**) and ventral (**b**) view of the medulla. **a** Location of the cranial nuclei 5 N, 7 N, 10 N, and 12 N are shown. **b** The location of 7 N and blood vessels, at the ventral surface of the brainstem, are used as reference-points ($N = 17$). **c** Experimental set-up with brainstem sectioned transversely for easy electrode access for micro-electrical stimulation (mES). Drawing adapted from Bouvier et al.[39], Copyright (2015), with permission from Elsevier. **d-g** mES-effective sites that evoke locomotor-like activity. Left panel—TdTomato (red) and Nissl (blue) staining. Right panel—heat map with effective sites of stimulation ($N = 17$). Source data are provided as a Source data file. **h** Schematic of split bath. Drawing adapted from Bouvier et al.[39], Copyright (2015), with permission from Elsevier. Glutamatergic neurotransmission blocked in the brainstem bath with 4 mM kynurenic acid (KA), eliminating the action of collateral pathways from glutamatergic brainstem neurons (green) to other descending neurons (black). **i-l** mES in the LPGi/CVL area after KA application. Source data are provided as a Source data file. **m** Representative example of mES-evoked locomotor activity. **n** Time-frequency plots (right panels) and circular plots (left panels) with left–right and flexor–extensor (contralateral to the stimulation site) activity. The color-coding denotes the coherence between the two signals and the vector field shows phase delay (also indicated in the circular plots, right = 0°, down = 90°) ($N = 6$ animals

indicated by the blue dots). Rayleigh test for left–right and flexor–extensor coupling are both significant ($p = 0.0061$ and $p = 0.0073$, respectively; two-sided). bef: before; af: after. Source data are provided as a Source data file. **o** Increased stimulation frequency leads to significant increase in the frequency of locomotor activities (one-way ANOVA, *$p = 0.022$) ($N = 5$ animals). Box-whisker plot shows median (middle line), 25th, 75th percentile and maximal and minimal value. Source data are provided as a Source data file. Scale bars (in μm): **b**: 500, **d–g**: 500. Abbreviations used in all figures: 10 N dorsal motor nucleus of vagus, 12 N hypoglossal nucleus, 12Nr/12Nc rostral/caudal hypoglossal nucleus, 12n hypoglossal nerve, 5 N motor-trigeminal nucleus, 7 N/n facial nucleus/nerve, AmbC ambiguus nucleus, Cb cerebellum, CVL caudal ventrolateral reticular nucleus, DMTg dorsomedial tegmental area, EVN efferent vestibular nucleus, Gi gigantocellular reticular nucleus, GiV the ventral part of the gigantocellular nucleus, IC inferior colliculus, IO inferior olive, LPGi the lateral paragigantocellular nucleus, LRt lateral reticular nucleus, MdD dorsal medullary reticular nucleus, MdV medullary reticular formation ventral part, MVe medial vestibular nucleus, PnC caudal pontine reticular nucleus, Pr5 principle sensory trigeminal nucleus, rGi/cGi rostral/caudal gigantocellular reticular nucleus, RMg raphe magnus nucleus, SOC superior olivary complex, Sp5 spinal trigeminal nucleus, sp5 spinal trigeminal tract.

in the LPGi/CVL area (the dotted square in Fig. 2d, $n = 7$, $N = 2$). In average, we found 38 CTB$^+$ neurons ($N = 2$, $n = 7$) per section. The CTB$^+$ neurons were clearly colocalized with YFP-positive Vglut2$^+$ neurons (34 %; Fig. 2f) and serotonergic neurons—as visualized with immunochemical staining for tryptophane hydroxylase (TPH) (14%; Fig. 2e). Because of the faint expression of YFP, it is likely that the CTB$^+$ Vglut2$^+$ colocalization is higher than detected. However, these results confirm that glutamatergic and serotonergic reticulospinal neurons are located in the LPGi/CVL area[25,42], the effective site for initiating locomotor-like activity in the newborn mice.

We then set out to examine whether activation of the serotonergic neurons in the LPGi exclusively may evoke locomotor-like activity using optogenetic stimulation in SERT$^{Cre}$;R26R$^{ChR2}$ preparations (Fig. 2h and Supplementary Fig. 4). In all experiments ($N = 6$ with aCSF in the brainstem; $N = 2$ with KA in the brainstem), optical stimulation of SERT$^+$ neurons in the LPGi area was unable to evoke rhythmic motor activity (Fig. 2i). Optical stimulation failed to evoke ventral root activity in $N = 6$ of 8 preparations, while in $N = 2$ of 8 preparations optical stimulation evoked modest tonic motor activity. Moreover, optical stimulation of SERT$^+$ neurons did not change the frequency or amplitude of locomotor activity evoked by mES alone ($N = 7$ out of 8, Supplementary Fig. 4c). These results strongly indicate that serotonergic neurons in LPGi do not initiate locomotion and cause little obvious modulation of the rhythmic activity.

We next asked whether activation of glutamatergic neurons in LPGi/CVL could induce locomotor-like activity. Optical stimulation of Vglut2$^+$ neurons in LPGi/CVL in Vglut2$^{Cre}$;R26R$^{ChR2}$ preparation (Fig. 2j) consistently evoked locomotor-like activity (Fig. 2k, $N = 5$). As with electrical stimulation, the optically induced locomotor-like activity persisted after blocking excitatory synaptic transmission in the brainstem pool (Fig. 2l, n, $N = 8$). The Vglut2-evoked locomotor activity exhibited similar frequencies ($0.84 \pm 0.03$ Hz) as the electrically evoked locomotion (Fig. 2o; $0.79 \pm 0.04$ Hz, stimulation frequency: 5–10 Hz) with left-right and flexor-extensor alternation (Fig. 2m, $N = 6$; phase value: 0.42 for both R/L and F/E; $r$: 0.88 (R/L) and 0.75 (F/E)).

Together, these results indicate that the micro-electrically defined LPGi/CVL area contains glutamatergic neurons that send a direct locomotor initiating signal to the lumbar spinal cord which does not depend on collateral activation in the brainstem or upper spinal cord. With these results in hand, we are now able to use the in vitro preparation to investigate how the tonic, non-rhythmic command signal leads to the initiation of rhythmic locomotor-like activity in the spinal cord.

## Transformation of tonic LPGi/CVL descending drive into rhythmicity

We first asked: which lumbar spinal locomotor networks are recruited by LPGi/CVL? To answer this question, we set out to perform large-scale Ca$^{2+}$ imaging of glutamatergic neurons from the lumbar spinal cord while stimulating the brainstem locomotor initiating area. We first focused on glutamatergic neurons because they are directly involved in the generation of rhythmic locomotor activity[2,13,27,43].

We recorded Ca$^{2+}$ signals from glutamatergic neurons by crossing Vglut2$^{Cre}$ mice with mice expressing the Ca$^{2+}$ indicator GCaMP6f (R26R$^{GCaMP6f}$). Vglut2 is expressed in most glutamatergic interneurons in the mouse spinal cord. By facing the transverse section of the cord to the microscope objective (×10, Fig. 3a), we directly visualized Ca$^{2+}$ signals in the entire transverse section of the cord (lumbar segment 3/4) before and during unilateral micro-electrical LPGi/CVL stimulation (Fig. 3a). We imaged lumbar segment 3, to maintain the most rhythmogenic part (L1-L2) of the spinal cord intact[41,44]. KA was applied to the brainstem pool to isolate the direct command on the spinal cord (Fig. 3a, left panel). Ventral root activity of left and right L2 segments was recorded to monitor the mES-evoked responses (Fig. 3a, c). Even though only three segments of the lumbar cord are preserved, electrical stimulation reliably evoke rhythmic activity with left and right alternation in L2 ventral roots.

We used 500 grid-created regions of interest (grid-ROIs) that tiled the entire transverse section of the spinal cord (Fig. 3a, right panel) and extracted the Ca$^{2+}$ signal (fluorescent transient, $\Delta F$) of the individual ROIs. We normalized the fluorescent transient of the individual grid-ROIs to the maximum value of all the grid-ROIs. We, then, established activity maps of all the grid-ROIs (Fig. 3d) for the entire stimulus duration, which generated a continuous visualization of network dynamics evoked by the LPGi/CVL descending command activity (Fig. 3e).

During LPGi/CVL stimulation, both the Vglut2$^+$ Ca$^{2+}$ signals and the ventral root activities showed two phases of activation: the *initiation phase* characterized by tonic activity and the *rhythmic phase* where the locomotor activity is established and sustained (Fig. 3c, e–g). The initiation phase lasted $3.74 \pm 0.41$ s. From the beginning of the initiation and throughout it, there was a strong tonic Ca$^{2+}$ signal in the ipsilateral lateral funiculus (e.g., Area 1 in Fig. 3d, e, g), an area containing reticulospinal tracts from LPGi[25] confirming that the brainstem stimulation recruits glutamatergic neurons in LPGi/CVL projecting to the spinal cord. In fact, approximately 100 ms (Frame #1 in Supplementary Fig. 5, $N = 7$) after the onset of the stimulation tonic activity already appeared in the ipsilateral cord with the earliest and strongest

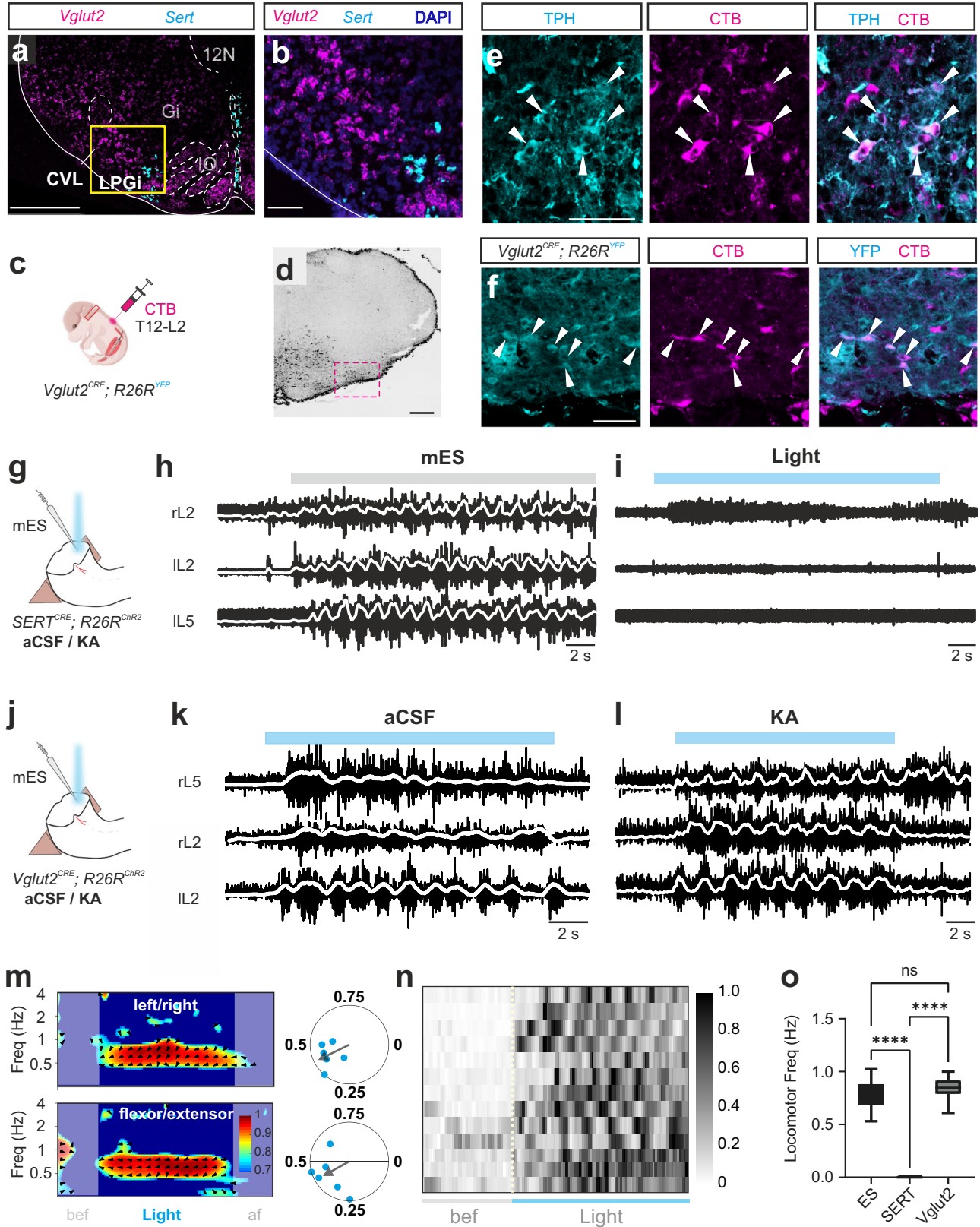

activity in the most medial part of the intermediate region of the cord (e.g., Area 3 in Fig. 3d, e, g, for laminar organization see Fig. 3b, Area 3 in Supplementary Fig. 5). Ca²⁺ activity appeared in the ventrolateral part of the cord with a slower onset (4.23 ± 0.95 s, N = 6) and lower intensity (e.g., Area 2 in Fig. 3d, e, g, Area 2 in Supplementary Fig. 5).

The rhythmic phase was characterized by the abrupt manifestation of rhythmic activity in the previously tonically active ipsilateral

medial laminae VII–VIII (Area 3) and ipsilateral ventrolateral area (Area 2). Rhythmicity also appeared—although weaker in intensity—in the contralateral ventral lamina VII (Area 4). The Ca²⁺ activity and the number of activated grid-ROIs are larger in the ipsilateral than the contralateral spinal cord in the initiation phase (n = 14, N = 7, Fig. 3f), confirming the asymmetrical nature of the descending signals. These results demonstrate that the unilateral LPGi/CVL command is received

**Fig. 2 | Glutamatergic, but not serotonergic, reticulospinal neurons in the LPGi/CVL interface directly with the executing spinal circuits to generate locomotor activity. a, b** Transverse section of brainstem showing RNAscope® in situ hybridization that targets Vglu2+ and Sert+ neurons at the level of rostral 12 N (*n* = 10 sections, *N* = 2 animals). **c–f** Retrograde labeling from the spinal cord of glutamatergic and serotonergic reticulospinal neurons in the LPGi/CVL area. **c** Bilateral injections of Cholera Toxin B (CTB) in the spinal cord in *Vglut2^{Cre};R26R^{YFP}* neonatal mice. **d** CTB staining at the level of rostral 12 N. The dashed square indicates the site of LPGi/CVL descending neurons (*n* = 7 sections, *N* = 2 animals). **e, f** Transverse sections stained for Vglut2+ cells (YFP), TPH (serotonergic neurons), and CTB. (*n* = 7 sections, *N* = 2 animals). **g** Experimental set-up for the *SERT^{Cre};R26R^{ChR2}* mice. Drawing adapted from Bouvier et al.[39], Copyright (2015), with permission from Elsevier. **h, i** Light activation of the serotonergic neurons at the mES-effective site did not evoke rhythmic ventral root (*N* = 6/2 animals for aCSF/KA condition). **j** Experimental design for the *Vglut2^{Cre};R26R^{ChR2}* mice. Drawing adapted from Bouvier et al.[39], Copyright (2015), with permission from Elsevier. **k, l** Light stimulation of glutamatergic neurons produced locomotor activity, under aCSF (**k**, *N* = 5 animals) and KA (**l**, *N* = 8 animals). **m** Coordination between left–right (top) and flexor–extensor (contralateral to the stimulation site, bottom) activity. *N* = 6 animals (indicated by the blue dots) for the circular plots. Rayleigh test for left–right and flexor–extensor coupling were both significant (two-sided; *p* = 0.0040 and *p* = 0.0254, respectively). Source data are provided as a Source data file. **n** Individual trials plotted to show the robustness of the locomotor-like activity evoked by optogenetic activation of glutamatergic LPGi/CVL neurons. The activity is represented as intensity plots from 0 to 1 (maximum activity of individual trials) (*n* = 17 trials in 7 animals). **o** Frequency of locomotor activities evoked by mES, light stimulation in *SERT^{Cre};R26R^{ChR2}* and *Vglut2^{Cre};R26R^{ChR2}* mice (*N* = 10, 8, 13 animals for ES, SERT, and Vglut2 conditions, respectively; one-way ANOVA, *p* = 0.000000000000000393). Post hoc analysis shows no significant difference between mES and LS in Vglut2 mice (*p* = 0.5476). Box-whisker plot shows median (middle line), 25th, 75th percentile and maximal and minimal value. Source data are provided as a Source data file. Scale bars (in μm): **a**: 500; **b**: 100; **d**: 500; **e, f**: 50.

by excitatory spinal modules that leads to the expression of: first a tonic initiation phase during which excitatory modules in the spinal cord are recruited to facilitate rhythm generation, and then in a rhythmic phase in which the modules are active to drive the locomotor-like activity.

## Identifying the immediate executor module of the LPGi/CVL command

Next, we set out to identify the excitatory module that directly receives the LPGi/CVL command, which is the logical target for the command execution. For this, first, we confirmed the LPGi/CVL entry location in the spinal cord by blocking the glutamatergic synaptic transmission in the spinal cord with KA application (Fig. 4a) in the *Vglut2^{Cre};R26R^{GCaMP6f}* preparations Indeed, the Ca²⁺ activity was only observed in the lateral funiculus of the ipsilateral cord, with little activity in the ventromedial funiculus (Fig. 4b, h, k, *N* = 5). These areas, therefore, contain descending fibers of LPGi/CVL neurons.

To identify the functional module that is directly targeted by the LPGi/CVL neurons, we perfused the lumbar spinal cord with 1 mM mephenesin, instead of KA (Fig. 4c). Mephenesin has been reported to attenuate and/or block polysynaptic transmission in the rodent spinal cord with less or no effect on the monosynaptic transmission[33,45–49]. We confirmed this in our preparation by recording ventral root activity during dorsal root stimulation, which showed the preservation of monosynaptic and blockage of polysynaptic responses under mephenesin (Supplementary Fig. 6a, b, *N* = 2). Mephenesin also blocked polysynaptic motor neuron responses after trains of stimulation of LPGi/CVL (Supplementary Fig. 6c, d, *N* = 12), and blocked the LPGi/CVL evoked rhythmic motor activity (Supplementary Fig. 6e–g, *N* = 12). Therefore, the application of mephenesin allowed us to visualize predominant monosynaptic responses to LPGi/CVL stimulation. We found that, except for the entry area, a predominantly ipsilateral cluster—located in medial laminae VII-VIII—was activated by LPGi/CVL stimulation (Fig. 4d, i, l, *N* = 5). This Vglut2+ cluster, which we call the immediate executor module (iEM) since logically it executes the LPGi/CVL command, was active very early. The activity was observed within the first frame of imaging (0–100 ms) after the onset of the stimulation (Supplementary Fig. 7) and remained active throughout the stimulation period: first, the activity was tonically active in the initiation phase, and then, immediately rhythmic in the sustained rhythmic phase.

Finally, under aCSF perfusion, with intact synaptic transmission in the cord (Fig. 4e), Ca²⁺ activity appeared sequentially in the LPGi/CVL command and then in the iEM (Fig. 4f), followed by the emergence of a new cluster located in the ventral part of lamina VII (highly activated area at 4 s in Fig. 4f; Fig. 4j, m, *N* = 5), close to or in the motor neuron area. We name this cluster the premotor module (prM).

The close proximity of the premotor module to the motor neuron pool and the fact that Vglut2 has been shown to be expressed weakly in motor neurons[50] raise the possibility that at least part of the signal in the prM module could be from motor neurons. To further qualify this conjecture, we performed Ca²⁺ imaging in *Vglut2^{Cre}; R26R^{GCaMP6f}* mice while we antidromically activated motor neurons by ventral root stimulation (4 Hz, pulse duration 200 μs, 10 s, 120–300 μA) on one side of the cord with simultaneous recording of the ventral root activity on the other side of the cord at the same segmental level. Under aCSF perfusion, ventral root stimulation might evoke bilateral rhythmic activity[51] because of release of glutamate from central motor neuron collaterals[50,51]. As an indication that the ventral root stimulation indeed activates the motor neurons on the stimulated side, we found activity in the contralateral ventral root. A clear Ca²⁺ signal was present in the intermediate area around the central canal with a weaker signal close to or in the motor neuron pools (Supplementary Fig. 8a–d, *N* = 3). When nicotinic receptors and glutamatergic receptors were blocked—thereby isolating the antidromic stimulation to the motor neuron pool (by removing any effect of central motor neuron collaterals on spinal circuits)—there was no calcium signal in the intermediate area and only a very weak calcium signal in the stimulated motor neuron pool (Supplementary Fig. 8e–i, *N* = 3). These data demonstrate that while motor neurons may give a weak ventrally located calcium signal in *Vglut2^{Cre}; R26R^{GCaMP6f}*—the contribution of this signal is minor and negligible to what we call the premotor module (prM). We also performed calcium imaging in *ChAT^{Cre}; R26R^{GCaMP6f}* mice. There was no calcium signal of the ChAT+ neurons under mephenesin in the spinal cord (Supplementary Fig. 8j, k) or rhythm in the ventral roots (Supplementary Fig. 6c, d) during LPGi/CVL stimulation. These data suggest that the LPGi/CVL stimulation does not activate the motor neurons directly. Under aCSF perfusion, there was clear activity in ventrally located bilateral clusters that was much stronger than that evoked by ventral root stimulation in the *Vglut2^{Cre}; R26R^{GCaMP6f}* mice (Fig. 4m). All together these data demonstrate that the activity in the premotor module (prM) originates predominantly from Vglut2 non-motor neurons.

In summary, our results indicate a specific spatiotemporal activation sequence in the initiation phase (Fig. 4g): the LPGi/CVL command travels from the lateral funiculus to the iEM in laminae VII-VIII, for rhythm generation, and then to the prM in the ventrolateral part of the spinal cord (Fig. 4k–m).

## Oscillatory profiles of the excitatory modules

The next question we asked is: in what way do the excitatory modules implement the locomotor activity? For this, we examined Ca²⁺activity during the rhythmic sustained phase (Fig. 5a, b). We quantified the rhythmic activity of individual grid-ROIs by first performing an

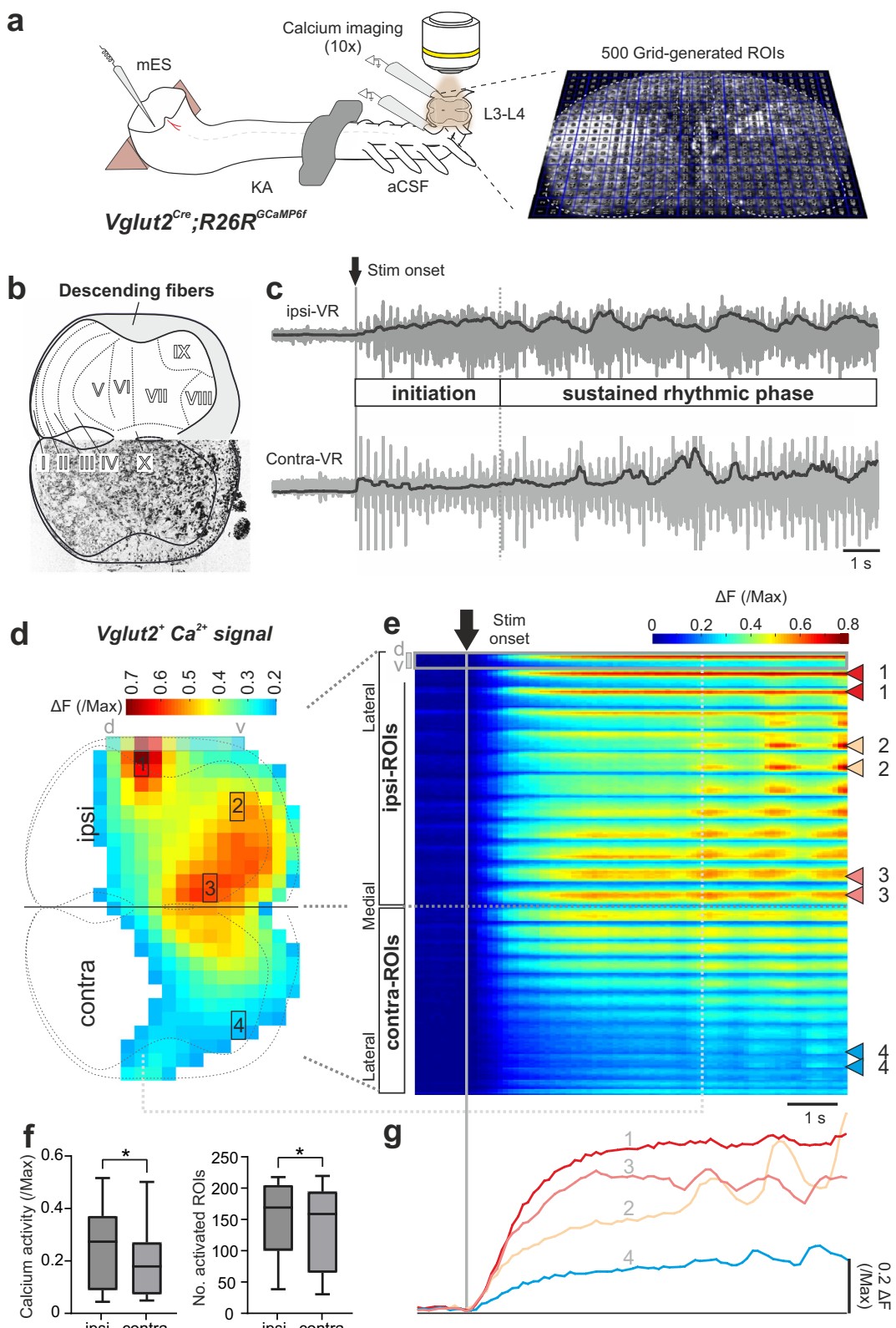

autocorrelation analysis (Fig. 5c). This yielded an oscillatory index—the value of the peak to the trough correlation coefficient—for each ROI (for details, see Methods), allowing us to assess the consistency and strength of oscillatory activity at discrete laminar positions across the spinal cord (Fig. 5d).

When oscillatory ROIs were tiled, we found that the averaged oscillatory strength and coverage were similar on the ipsilateral and contralateral sides (Fig. 5e, f), indicating bilateral participation of the spinal networks. The most ventral part of iEM exhibited comparatively strong and steady oscillation (Fig. 5d). The ipsilateral prM had the strongest intensity of oscillations of all areas. The contralateral prM, on other hand, exhibited weaker oscillation. These results show that the spinal locomotor circuitry recruited by the LPGi/CVL command is highly organized and that diverse oscillation strength underlines the complexity of the network operation in a modular organization.

**Fig. 3 | LPGi/CVL drive is transformed from an initial tonic phase to a state of sustained rhythmic activity. a** Left, experimental set-up for Ca$^{2+}$ imaging in *Vglut2$^{Cre}$;R26R$^{GCaMP6f}$* mice. Drawing adapted from Bouvier et al.[39], Copyright (2015), with permission from Elsevier. Ca$^{2+}$ imaging performed from the cut surface of the spinal cord at the level of L3/4 during LPGi/CVL stimulation. Right panel, frame of Ca$^{2+}$ signal (fluorescence transients, ΔF) at a random time point. The recording field is divided into 500 grid-generated ROIs. **b** Upper panel: cytoarchitecture of the spinal cord showing the laminar organization and areas of descending fibers of L3. Lower panel: transverse section of the spinal cord at the level of L3 stained for NeuroTrace. **c** Representative ventral root (VR) recordings during LPGi/CVL stimulation show two distinct phases: the tonic initiation phase and a rhythmic phase. **d**–**g** Ca$^{2+}$ activity of Vglut2$^+$ neurons in the two phases. **d** Activity map in the spinal cord 5 s after stimulation onset. Only the grid-ROIs with activity larger than the 0.2 maximal value were plotted. Specific areas are numbered from 1 to 4. **e** Heatmap of activity of individual grid-ROIs shown in **d** before and during stimulation. The

initiation and rhythmic phases are clearly seen. Numbered triangles correspond to numbered grid-ROIs in **d**. Each row represents the calcium activity for a grid-ROI across time. Dorsal (d)−ventral (v) stretching rows in **d** are represented by time-series columns in **e**. In each column, the ROIs from the dorsal to the ventral stretching row (**d**) is represented with dorsal upwards and ventral downwards. See Supplementary Fig. 5 for details. **f** The intensity of Ca$^{2+}$ activity (left) and the number of activated ROIs (right) are visibly larger in the ipsi- than in the contra-lateral spinal cord ($n = 14$ trials from $N = 7$ animals). Box-whisker plot shows median (middle line), 25th, 75th percentile and maximal and minimal value (one-sided two-way repeated measure ANOVA; ****$p = 0.00000000368$; post hoc analysis: $p = 0.0247$; and 0.0197 for Ca$^{2+}$ activity and the number of activated ROIs, respectively. *$p < 0.05$). Source data are provided as a Source data file. **g** Example of Ca$^{2+}$ traces for four different ROIs (1, 2, 3, 4 in **d**, **e**) with different activity patterns during stimulation. ipsi/contra: ipsilateral/contralateral.

## Real-time network dynamics of oscillatory excitatory modules

Next, in order to understand how the excitatory modules work together, we tracked how activity traveled across excitatory modules throughout the entire locomotor cycle. For this, we employed cross-correlation of all the 500 ROIs with the ipsilateral prM as the reference. An example of cross-correlation between individual excitatory modules is shown in Fig. 6a, revealing a specific temporal sequence. Then, we reconstructed the cross-correlation map of the entire transverse section of the spinal cord for the entire locomotor cycle (Fig. 6b). The tracking of the activity shows that it propagates from the ipsilateral cord (where the stimulation is applied) and then to the contralateral cord.

At $T = -0.2$ of the locomotor cycle (prior to the peak activity of ipsilateral prM by 20% cycle duration, Fig. 6b), the activity of neurons in the ipsilateral cord started to emerge, while its prM was still quiet. Among the early emerged clusters, the iEM started to exhibit high activity before the peak of ipsilateral prM ($T = 0$). The ipsilateral wave faded away at $T = 0.3$, when the contralateral cord activity started to emerge. The contralateral prM reached the peak at $T = 0.5$-$0.6$ and faded away at $T = 0.8$ (i.e., $T = -0.2$) when the ipsilateral network activity started to emerge again.

This traveling-wave pattern among the excitatory modules was seen in the averaged phase map ($N = 7$, Fig. 6c, d). The first cluster that started the cycle in the ipsilateral cord, i.e., ROIs with phase values from −0.2 to 0, was corresponding to iEM. Then the wave traveled to the ipsilateral prM as well as the contralateral spinal cord ($T = 0$-$0.2$). The contralateral prM reached a peak at $T = 0.4$-$0.6$.

Altogether, the Ca$^{2+}$ imaging of the excitatory spinal networks unravels when and where the descending LPGi/CVL command signal is translated into rhythmicity: it is received and "read" by the iEM−the immediate executor module−which is the first module to be rhythmic (Fig. 3e). The iEM then "passes" the rhythmicity to the prM, which is composed of premotor interneurons before it reaches motor neurons for behavioral execution.

## Inhibitory modules are distinct from excitatory ones

We next targeted the inhibitory neurons that play an essential role in patterning locomotor activity, including left-right and flexor-extensor alternation, to investigate their recruitment by the LPGi/CVL command. For this, we performed Ca$^{2+}$ imaging in *Vgat$^{Cre}$;R26R$^{GCaMP6f}$* mice ($N = 7$). LPGi/CVL stimulation promoted some Ca$^{2+}$ activity in the lateral funiculus of the ipsilateral cord that is present in both KA ($N = 3$; Supplementary Fig. 9a, b, i) and mephenesin ($N = 7$; Supplementary Fig. 9c, d, h). In four preparations under mephenesin, the LPGi/CVL stimulation activated a small cluster of inhibitory neurons close to the central canal in the intermediate spinal cord (Supplementary Fig. 9h). This cluster was not present in all preparations. Under aCSF, in the later part of the initiation phase, large bilateral areas in the ventral spinal cord were activated (Supplementary Fig. 9e–g). These results show

that the electrical stimulation, in addition to excitatory LPGi/CVL locomotor initiating fibers, activates some of the inhibitory descending fibers; an inhibitory cluster is directly activated by the LPGi/CVL.

The activity map during the rhythmic phase of the locomotor activity is shown in Supplementary Fig. 10a. The Ca$^{2+}$ activity of the inhibitory neurons exhibited a very different pattern from that of excitatory neurons (Fig. 4j). The activated areas and calcium activity of the ipsi- and contra-lateral cord were not significantly different (Supplementary Fig. 10c). Indeed, the activity was broadly distributed in laminae VII−VIII (Supplementary Fig. 10a, b)−locations for interneurons that secure flexor−extensor alternation[52−56] and left−right alternation[57−61]. The network dynamics analysis revealed that the ipsilateral inhibitory modules were activated in the first half of the locomotor cycle ($T = -0.2$ to $T = 0.3$), while the contralateral modules were activated in the latter half ($T = 0.3$ to $T = 0.8$) (Fig. 7a, b). These results indicate that inhibitory spinal neurons are more symmetrically activated after unilateral brainstem command and that the ipsi- and contra-lateral networks operate antagonistically to ensure left−right alternation.

## Discussion

The present study uncovers the functional chain of the locomotor command signal from the brainstem to the spinal cord and reveals the real-time operation of the excitatory and inhibitory spinal locomotor circuitry that is activated from this command signal. The study directly visualizes the modular operation of the spinal locomotor network at the macroscale.

In this study, we found that non-cell-specific micro-electrical stimulation of neurons in the LPGi/CVL area consistently evoked prolonged locomotor-like activity. Unconditional light-induced activation of MRF neurons including neurons located in the LPGi, the GiA, GiV, and Gi[25,62], in the mouse, however, was not successful in evoking locomotion. The reasons for this difference from previous experiments may be explained in several ways. First, the in vitro approach enabled us to visually find effective spots from the rostral cut end of the brainstem which brought the electrode to the point of stimulation. Secondly, by slicing stepwise the brainstem from the rostral part, we disconnected the effective area from the rostral brainstem and brain areas removing any interference from structures not stimulated directly. Because of the in vitro condition, we were also able to demonstrate−by blocking the glutamatergic collateral activity in the brainstem, and upper spinal cord−that the descending signal originated in the stimulated site and not from collateral branching in the brainstem. This later condition is not yet possible to establish in vivo with current technology. Indeed, we show that non-cell-specific micro-electrical stimulation of the lateral medulla along the rostrocaudal levels evokes locomotor bursts, with a locomotor-like pattern. These areas form a strip, which corresponds to the previously identified pontomedullary locomotor strip, a descending tract that

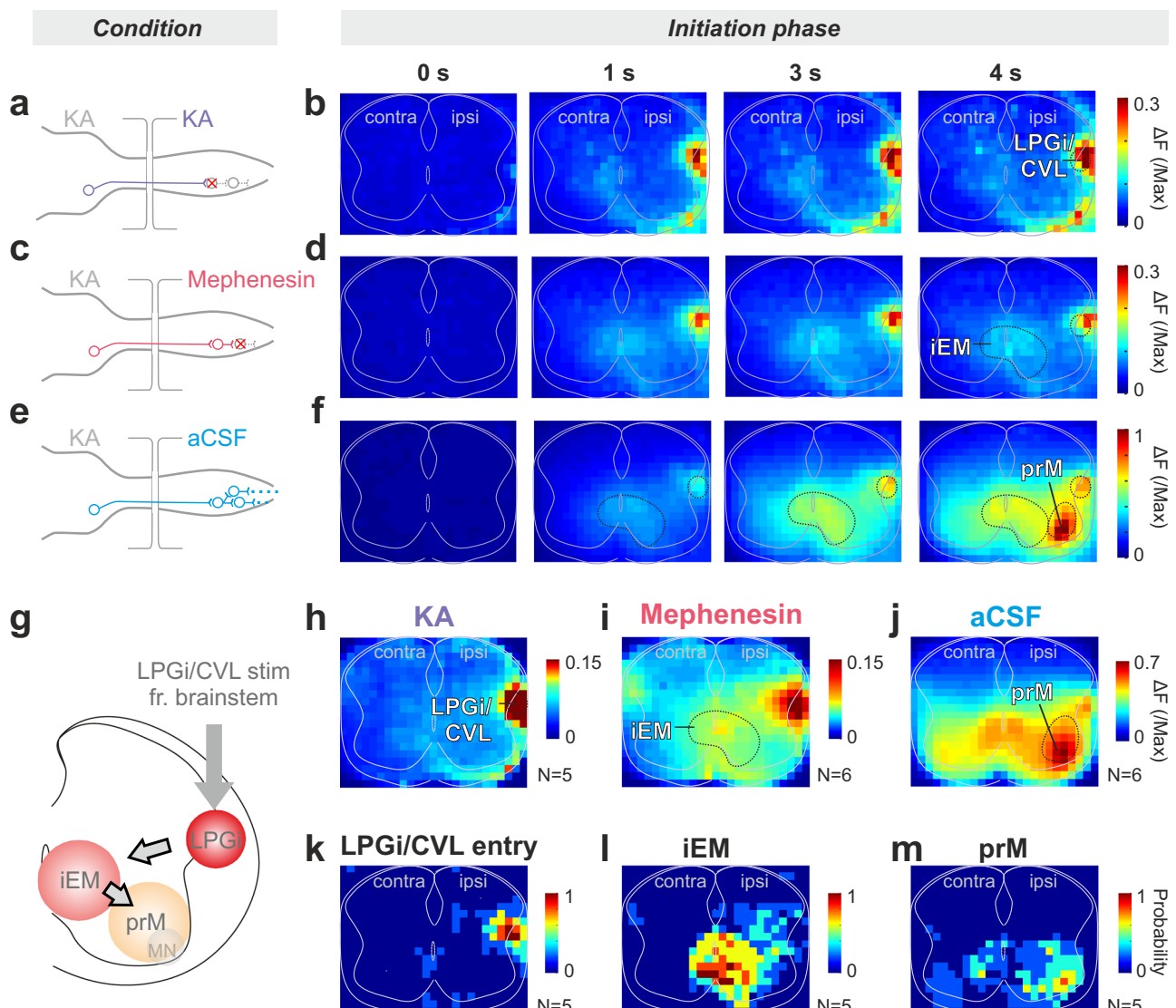

**Fig. 4 | Deconstruction of activity in the tonic initiation phase identifies the excitatory module that may directly execute the LPGi/CVL command.**
**a**–**f** Pharmacological investigation in a representative preparation that identifies different glutamatergic clusters with specific functions. **a**, **c**, **e** Schematics of the experiments and putative neuronal connection with each drug. **b**, **d**, **f** Activity maps of different time points after onset of the stimulation. Under KA (**a**, **b**), glutamatergic transmission in the spinal cord is blocked, thereby revealing the location of the glutamatergic descending signals, the LPGi/CVL entry. Under mephenesin (**c**, **d**), the polysynaptic recruitment by descending drives is blocked, thereby

revealing the monosynaptic glutamatergic downstream target, the iEM (see "Results"). Under the aCSF condition (**e**, **f**), when the synaptic activity is intact, another cluster, prM, emerges 4 s after stimulation. **g** Schematic showing the temporal activation sequence of the three glutamatergic clusters. **h**–**j** Averaged maps of all preparations under the KA (**h**), mephenesin (**i**), and aCSF (**j**) condition at 4 s after stimulation onset. Source data are provided as a Source data file. **k**–**m** Probability of LPGi/CVL entry (**k**), iEM (**l**), and prM (**m**) among animals. Source data are provided as a Source data file. Abbreviation for Figs. 4–6: iEM immediate executor module, prM premotor module, MN motor neuron.

evokes locomotion in cats[15,17,63–66]. However, the electrically induced locomotor-like bursts were absent or largely diminished after blocking the collateral glutamatergic activity in the brainstem and upper spinal cord, suggesting that these areas mediate their effect through collaterals in the brainstem, possibly acting through the LPGi/CVL or through other polysynaptic relays in brainstem or upper spinal cord. Moreover, in the lamprey, sensory-triggered swimming bouts are mediated through brainstem neurons activated by trigeminal sensory stimulation nucleus that projects to reticulospinal neurons that mediate the locomotor command[11,67].

Earlier experiments using micro electrical stimulation of the MRF have shown that locomotion can be initiated from Gi in the adult cat[7,13,15–17,19,68,69]. However, activation of Gi in adult or neonatal mice fails to initiate locomotion[25,34,62] and instead induce turning and stopping of locomotion[39,40]. It is possible that these differences are

due to species difference or that a cell specific activation in the cat would reveal the contribution from the more lateral localized LPGi/CVL. The LPGi/CVL region is physically close to the parafacial respiratory group. Interestingly, lesion of the parafacial respiratory group suppressed the locomotor-induced accelerated respiration[70]. Further experiments are required to clarify the neuronal connection between these locomotor initiating and respiratory brainstem structures.

Using focused optogenetic stimulation, similar to electrical stimulation, we show that glutamatergic, but not serotonergic, neurons in the LPGi/CVL region can activate a direct command pathway for initiating locomotor-like activity in the neonatal mouse. This finding underscores that the glutamatergic neurons in LPGi/CVL form a command pathway. Glutamatergic neurons in LPGi have also been shown to initiate locomotion in the adult mouse[25] and to control the speed of

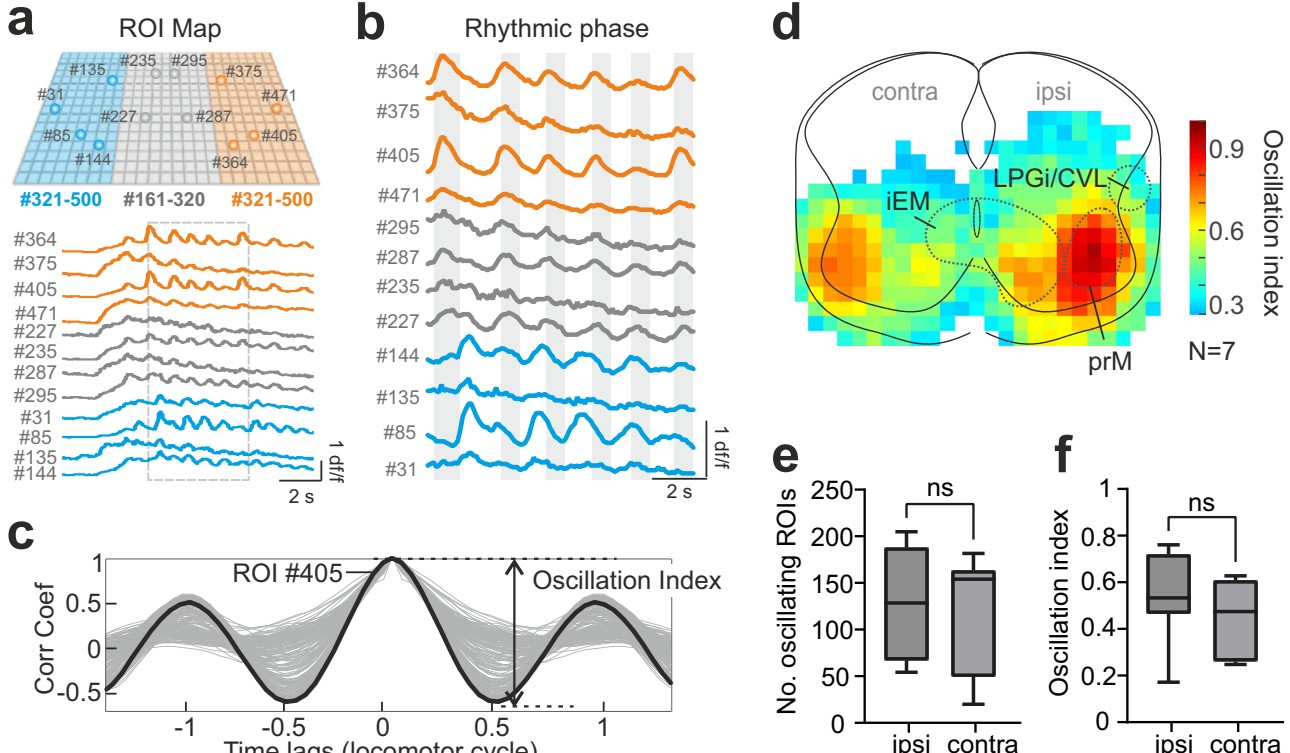

**Fig. 5 | Oscillatory profiles of the excitatory modules reveal the modular operation during the rhythmic sustained phase. a, b** Examples of Ca²⁺ signals extracted from 500 individual ROIs (orange, gray, or blue circles in upper panels of **a**) are shown in the lower panel of **a** for the entire time span and **b** for the rhythmic phase, respectively. **c** Autocorrelation analysis of the Ca²⁺ signal in individual ROIs is used to reveal the oscillation strength. Light/dark gray line indicates traces for individual ROIs/ROI #405. **d** Oscillation map with the spatial relationship of ROIs with different degrees of oscillation. The color coding denotes the activation/oscillation strength of the designated location. Source data are provided as a Source data file. **e, f** The number of ROIs and oscillation index of the ipsilateral and contralateral spinal cord is not significantly different (two-way repeated measure ANOVA, $p = 0.467$; $n = 13$ trials from 7 animals). Box-whisker plots show median (middle line), 25th, 75th percentile and maximal and minimal value. Source data are provided as a Source data file.

locomotion in a frequency dependent way[25]. We reproduce the frequency dependency in this area with electrical stimulation although we were not able to do so with optical stimulation because we always used long square pulses. Nevertheless, the convergence of LPGi in neonatal and adult mice suggests that LPGi is ontogenetically preserved for locomotor initiation. However, we have also found that the locomotor-initiating area includes CVL in the neonatal mouse. This apparent difference between neonatal and adult mice may reflect developmental changes. Nevertheless, because of the intact glutamatergic transmission in the brainstem in adult mouse experiments it seems difficult to completely exclude a contribution from CVL even though it was not stimulated directly.

Stimulating the serotonergic descending neurons in and close to LPGi did not evoke locomotion but their activity might act on the spinal cord to increase the excitability of the locomotor networks and have an indirect locomotor supporting effect. In this study, we, therefore, used electrical stimulation of LPGi/CVL to activate the command pathway because it allowed us to perform simultaneous Ca²⁺ imaging from the cord while stimulating the effective locomotor-promoting brainstem area[32,33]. We cannot exclude that the electrical stimulation also activates serotonergic fibers. Notably, optical stimulation of serotonergic fibers did not change the electrically induced locomotor activity suggesting that their contribution to the overall descending signal is low. It was clear, however, that electrical stimulation activated some descending inhibitory fibers in the lateral funiculus. Nevertheless, inhibitory synapses should not cause a Ca²⁺ signal, unless they cause a depolarizing postsynaptic event. However, there is a shift from depolarizing to hyperpolarizing inhibition around birth in the rodent embryos[71] which is further enhanced by the tonic

depolarization in the initiation phase. Therefore, stimulation of these inhibitory fibers will not contaminate the excitatory calcium signal we observe and thereby our conclusions. Another possibility is that the inhibitory drive will suppress some neurons in the cord. However, we always saw a clear tonic excitation in the initiation – so that possibility is less likely.

Different from previous studies focusing on identifying Ca²⁺ signals of individual cells[32,72–75], our study used broad-scale imaging and analyzed the Ca²⁺ activity of ROIs that are distributed in the entire transverse section of the spinal cord. Even though this method does not highlight individual cellular activity, it allows us to reconstruct the network operation at a macro-scale, by identifying modules with neuron assemblies operating concurrently. We focused in the first place on the excitatory neurons because glutamatergic neurons are known to be rhythm-generating neurons[2,27] and might be a logical first target of the descending locomotor command[13]. We targeted all excitatory neurons to get a broad view of the recruitment rather than single classes of glutamatergic neurons[2,27,28,43]. It is possible that the origin of the recorded Ca²⁺ activity may not only be cell bodies per se but also a concentration of synaptic terminals. Notable the glutamatergic descending axonal bundles show a clear Ca²⁺ signal when activated suggesting the presence of Ca²⁺ channels in axons[76,77]. Indication of the latter comes from the observation that the descending pathways in the lateral funiculus displayed a clear Ca²⁺ signal during LPGi/CVL stimulation. Moreover, what appears as terminals from descending neurons gave a weak signal in the intermediate area when all glutamatergic synaptic activity is blocked in the cord. In all cases, the Ca²⁺ imaging seems to track the neuronal activity as it travels through the spinal cord.

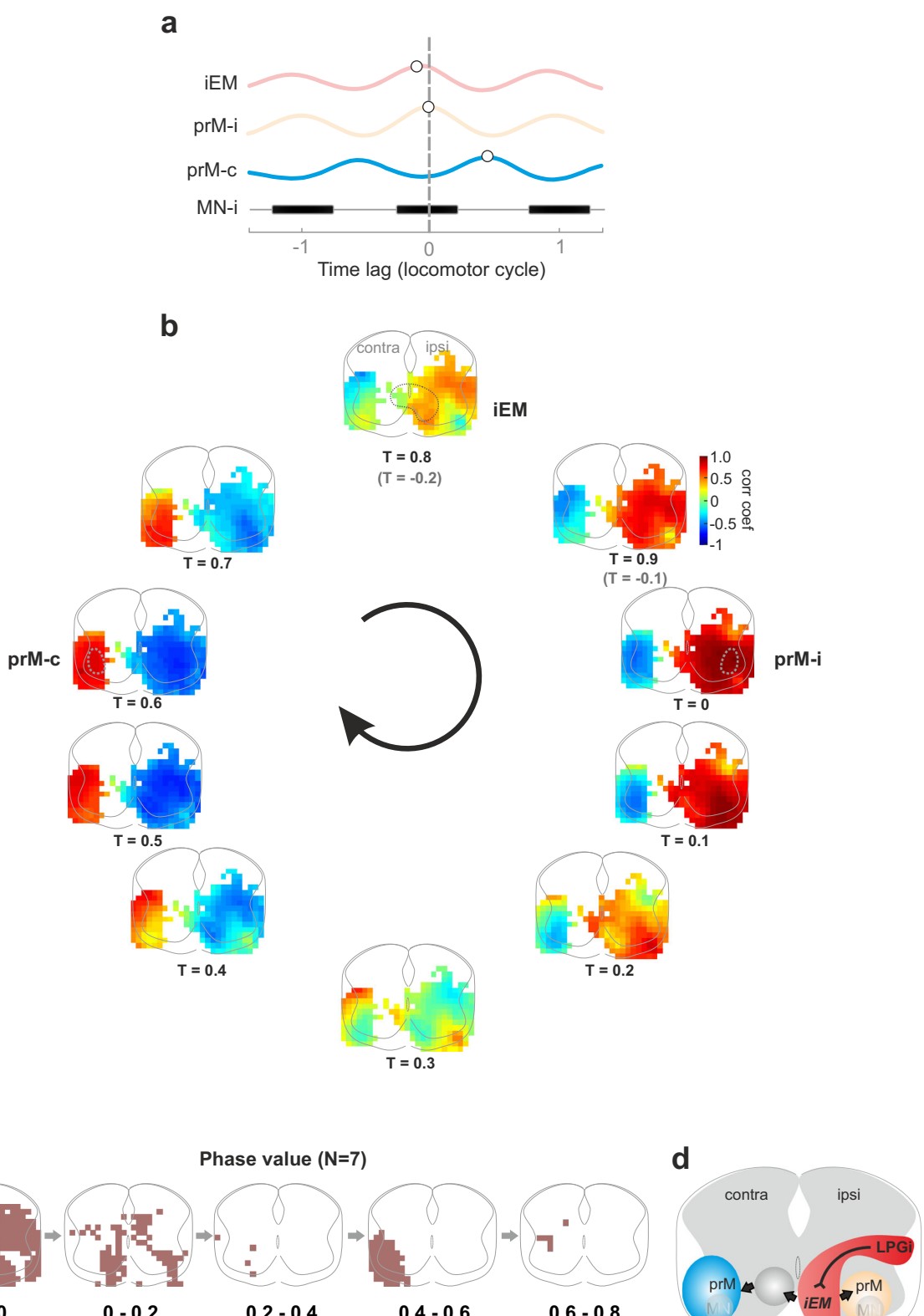

**Fig. 6 | Real-time network imaging unravels the hierarchy of excitatory modules that operate together to generate rhythmic locomotor activity. a** Cross-correlation showing the timing of the peak activity of the representative ROIs of the three identified glutamatergic areas in relation to the activity in the motor neurons (schematically illustrated). The ipsilateral iEM oscillation precedes the ipsilateral premotor area, while the contralateral premotor area is in antiphase with the ipsilateral one. **b** Dynamics of network activity. Frames of the cross-correlation maps of sequential time points. By tracking highly activated areas (warm-colored areas) in each frame, traveling of rhythmic areas in the spinal cord and across the locomotor cycle appears. **c** Phase map showing the timing of the peak activity of individual areas averaged among preparations. The arrows indicate the direction of wave propagation. Source data are provided as a Source data file. **d** Schematic drawing of the network operation of the identified glutamatergic clusters.

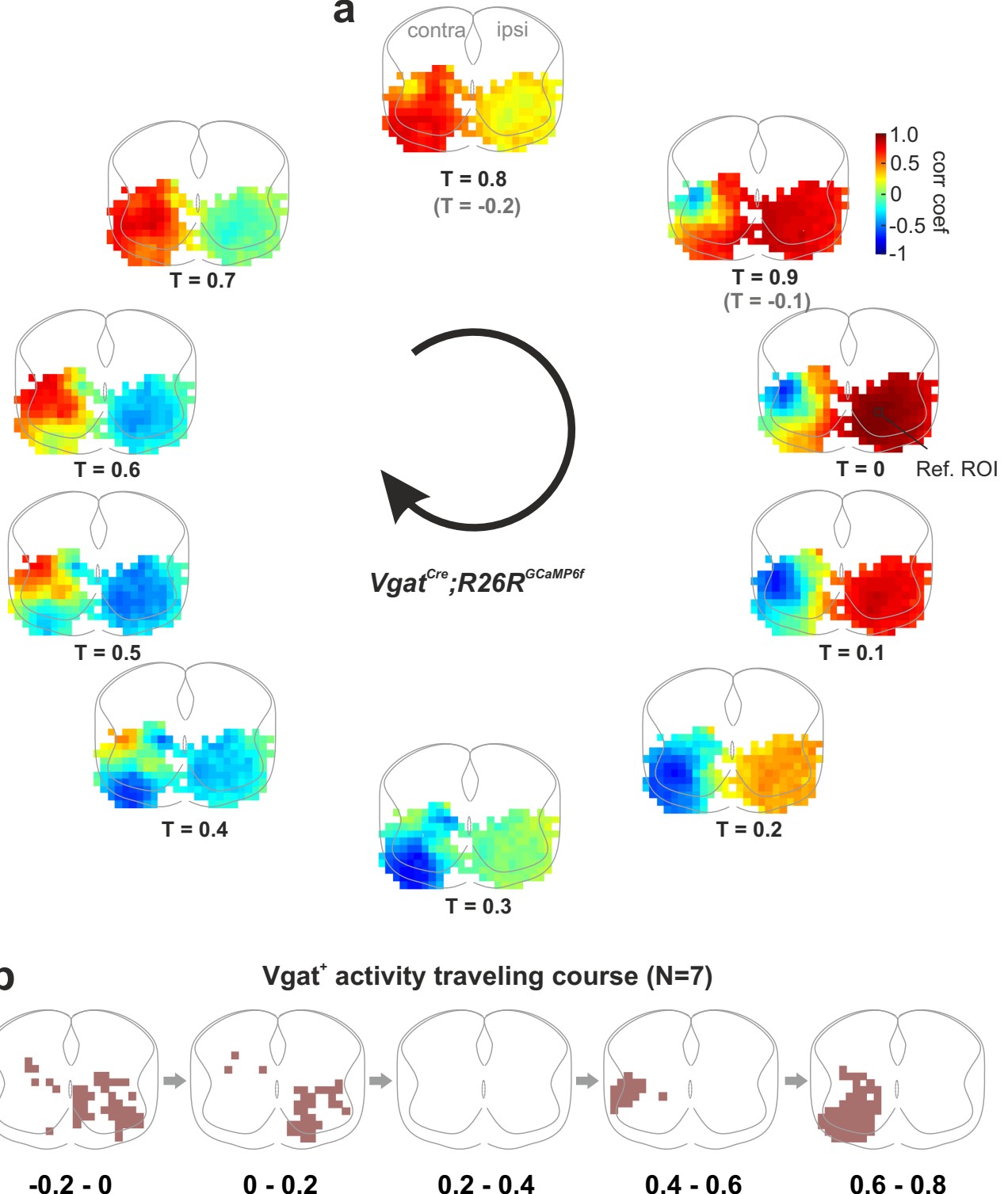

**Fig. 7 | Inhibitory modules of the spinal locomotor network are recruited symmetrically by the LPGi/CVL drive with strict left-right alternation.**
**a**, **b** Phase maps for inhibitory cells during locomotor activity evoked by LPGi/CVL drive. The arrows indicate the direction of wave propagation. Ref. ROI the reference ROI of the cross-correlation analysis. Source data are provided as a Source data file.

Many studies have used drug application or broad electrical stimulation to recruit spinal lumbar locomotor networks in the in vitro rodent spinal cord. These methods for initiating locomotor-like activity have been applied extensively when investigating the organization of the spinal locomotor circuitries. However, the immediate

descending brainstem command is presumably acting on specific neuronal populations. We performed a brainstem screening and located the only spot that sends specific locomotor commands to the spinal cord. With the stepwise pharmacological intervention in the spinal cord, in this study, we identify three glutamatergic modules in

the network: the LPGi/CVL axons in the lateral funiculus in the spinal cord, its monosynaptic recipient in the cord, which we call the immediate executor module (iEM) and a premotor module, prM, that is downstream of the executor module.

The LPGi/CVL entry module was tonically activated during stimulation, and in turn activates the iEM. iEM responds first to the descending LPGi/CVL command. Its activity is present after pharmacologically blocking or reducing the polysynaptic responses with mephenesin. iEM was tonically activated almost from the beginning of LPGi/CVL stimulation (the first frame of 100 ms after onset). This module continued to be highly activated in the tonic initiation phase. The time to onset for observing a brainstem-evoked $Ca^{2+}$ signal in the lumbar spinal cord is compatible to what has been observed before by others using $Ca^{2+}$ imaging of motor neurons in the in vitro neonatal mouse preparation[33]. The conduction velocity for reticulospinal neurons have been calculated to be ~0.5 m/s[78] in the newborn mouse which will give arrival times to the lumbar cord in around 35-40 ms, a shorter time than we were able to resolve with our 10 Hz sampling rate. However, in all cases, latencies for activation do not allow us to say anything about mono- or poly-synaptic connections from LPGi/CVL to the spinal locomotor target. But the stepwise pharmacological intervention in the spinal cord provides this information and highlight iEM as the primary target for the activation. iEM later became rhythmic at the beginning of the rhythmic sustained phase. The delay in conversion from tonic activity to rhythmicity is not unique to our preparation but is seen even in vivo with an intact nervous system. MLR activation often takes seconds to activate locomotion in mice[8,9,79,80] and cats[7,38] and spinal neurons present tonic activity before the emergence of rhythmic activity[81,82].

Interestingly, the iEM location is very similar to the location that Noga et al. identified in the adult cat with electrophysiological methods with a much faster time resolution than $Ca^{2+}$ imaging[83]. They showed that the earliest excitatory input from stimulation the locomotor promoting cuneiform nucleus is concentrated in lamina VII and, to a lesser extent, in lamina VIII with no sign of monosynaptic activation of the motor neuron as also seen in our study. Noga et al. did not reveal the rhythmicity of the neurons that receive the input and the traveling wave in the cord. Interestingly, in the cat the input signal was present in lamina VII throughout the lumbar spinal cord with a peak corresponding to L4-L6[2,83]. Here we only investigated the L3/L4 level and are therefore unable to know if the pattern is the exact same in more rostral (L1-L2) or caudal regions like L5-L6. We avoided imaging L1-L2 because this region is most rhythmogenic and cutting at this level often destroy the rhythm[29]. However, it is known that L3 and more caudal segments in the rodent spinal cord also can generate rhythm in isolation from L1-L2 (with drugs[37] and optogenetics[44]) which suggest a preservation of the rhythmogenic localization at least at the level of L3/L4.

When iEM becomes rhythmic, the premotor module also becomes rhythmic but with a distinct time delay to iEM. This sequence of activity indicates that the immediate executor module translates the LPGi/CVL descending command into rhythmicity, then feeds this rhythmicity to the premotor module, and in the end, activates motor neurons to generate the locomotor-like motor output. Since some motor neurons also express a weak Vglut2 signal[50], some of the activity in the prM is likely from motor neurons. However, our ventral root stimulation showed little motor neuron signal in Vglut2 mice. Moreover, the prM cluster in the Vglut2 mice is much stronger and encompasses a larger area than the ChAT cluster. These data strongly suggest a major contribution from Vglut2+ interneurons in the prM. In all cases, the visualization of Vglut2 $Ca^{2+}$ signal shows where and when the locomotor circuit in mammals is activated from the brainstem and its real-time dynamics. Instead of being scattered, the glutamatergic functional modules are highly organized physically in space and activated in a strict temporal sequence that is executed on the ipsilateral side of the cord before it is repeated on the contralateral side. A contributing reason for why we were able to visualize this sequence is that the descending signal was unilaterally activated and recruited the ipsilateral locomotor network first in the initiation phase and then propagated bilaterally in the maintenance phases.

The characteristics of the excitatory executor module—with its direct activation from locomotor command and primary oscillatory activity—are compatible with it being the core rhythm generating neurons in the spinal locomotor circuits. The sequential activation from one glutamatergic module to the next bears resemblance to the proposed layered organization of the mammalian locomotor network where the initial rhythm generating circuit is separated from excitatory premotor circuits[2,84]. The location of iEM corresponds to a region of the spinal cord that includes glutamatergic neurons that have been suggested—based on ablation and activation experiments—to be involved in rhythm generation in mice[83]. These populations include the glutamatergic population of non-V2a Shox2 neurons[2,26], the glutamatergic Hb9 neurons[28,85,86], as well as the recently identified ventral spinocerebellar tract neurons in mice[87]. It will be of interest in future experiments to determine activation of the existing identified rhythm-generating neurons from LPGi/CVL as well as activation of other functionally defined excitatory neuronal subpopulations (like the Chx10-V2a neurons[88–92]). It will also be of great interest to determine similar entry points in the zebrafish spinal cord where whole spinal cord imaging can be done more effortless than the mouse spinal cord[93–96]. Work on V2a interneurons in the adult zebrafish has already shown that excitatory V2a interneuron microcircuits in the spinal cord are recruited differentially to secure speed of locomotion and revealed the mechanism for the rhythm generated in these circuits[97–99].

The inhibitory components of the spinal locomotor network are considered to have a major role in patterning rhythmic motor neuron activity[2,3,27,71]. The network dynamics of the inhibitory neurons support this notion. We show a relatively symmetric inhibitory network on both sides of the cord. The ventrally located ipsilateral networks operate antagonistically with the contralateral networks, suggesting that some of these networks play a role in segmental left–right alternation. Commissural neurons involved in left–right alternation are localized in the medial laminae VII–VIII of the cord[57–61,100] corresponding to the strongly oscillating left–right reciprocal pattern in the inhibitory network dynamics map. Additionally, there must be inhibitory networks securing flexor–extensor patterning that reside on the same side of the cord. The oscillatory activity, seen in the laminae VII–VIII, is likely to represent such circuitries that may act on motor neurons in the same segment or motor neurons in more distant segments. It will require a more detailed analysis—than presented here—of genetically defined subpopulations of inhibitory neurons, e.g., V1 and V2b[55,56] to decode the real-time network operation for the inhibitory component of the spinal locomotor network.

The combined approach of this study—that enables us to selectively activate the brainstem command pathways while recording the system-wide and cell-specific activity pattern in the spinal cord with high spatial accuracy—required an in vitro approach. Such experiments cannot be performed in vivo or in adult animals. The question that arises is, therefore, if the organization of the nervous system in the neonatal mice is similar in adult mice, and if our findings generalize to adult mice. We do not know the answer to this question. The newborn rodents do not walk, but locomotor networks in the spinal cord can already at birth produce left–right and flexor–extensor alternation together with precise interlimb coordination[29] like in adult walking animals[37]. Moreover, neonatal rodents can generate physical stepping movement[101] as well as air-stepping movement[102]. The reticulospinal neurons also develop before birth and establish functional connections in the spinal cord in the neonate[103]. Recent experiments have also shown that brainstem command circuits controlling turn and stop function is similar in newborn and adult mice[39,40]. Here, we also clearly

demonstrate that the locus of the locomotor command neuron overlaps with that found in adult mice[25]. Thus, it is most likely that the basic circuitry from the brainstem to the spinal cord is structured at least in broad strokes in the same way and that our study captures the general features of the locomotor circuit organization across ontogeny in mice.

In conclusion, this study completes the functional chain between the brainstem command system and the spinal execution circuits that produce locomotion. While our work does not speak to the cellular mechanism of network function, it defines its phenomenology and provides an analytical framework to decode complex motor networks which produce locomotion.

## Methods

All animal experiments and procedures were approved by the local ethical committee, Stockholm's Norra Forsöksdjursnämnd (N42/16) and Dyreforsøgstilsynet (2017-15-0201-01246; 2017-15-0201-01172) in Denmark. These experiments were performed in accordance with European guidelines for the care and use of laboratory animals.

### Mice lines

The following transgenic lines of both sexes were used: $Vglut2^{Cre104}$, $Vgat^{Cre}$ (Jackson Stock 016962), $SERT^{Cre}$ (Jackson Stock 014554), $ChAT^{cre}$ (Jackson Stock 006410), $R26R^{GCaMP6f}$ (Jackson Stock 030328), $R26R^{Tdtomato}$ (Jackson Stock 007914), $R26R^{YFP}$ (Jackson Stock 006148), and $R26R^{ChR2-EYFP}$ (Jackson Stock 012569).

### Dissection

Mice of both sexes, 0–4 days old, were anesthetized in isoflurane and then eviscerated. The brainstem and the spinal cord were dissected free in cool (4 °C) low-$Ca^{2+}$ Ringer's solution that contained 111 mM NaCl, 3 mM KCl, 11 mM glucose, 25 mM NaHCO$_3$, 3.7 mM MgSO$_4$, 1.1 mM KH$_2$PO$_4$ and 0.25 mM CaCl$_2$. The solution was aerated with 95% O$_2$ and 5% CO$_2$. After dissection, the brainstem-spinal cord preparation was transferred to a recording chamber with normal Ringer's solution containing 111 mM NaCl, 6 mM KCl, 25 mM NaHCO$_3$, 1.25 mM MgSO$_4$, 1.1 mM KH$_2$PO$_4$, 11 mM glucose and 2.5 mM CaCl$_2$, aerated with 95% O$_2$ and 5% CO$_2$. Recordings were performed at room temperature (22–23 °C).

### Electrical identification of brainstem locomotor command region

To investigate the effective sites of brainstem stimulation that can evoke locomotor-like activity, brainstem-spinal cord preparations of $ChAT^{Cre};R26R^{TdTomato}$ mice were used since the cranial motor nuclei with different rostrocaudal levels are clearly marked. The brainstem was sectioned in the transverse plane at four different rostrocaudal levels with a fine scalpel. By placing the preparation ventral side up the specific pattern of vessels running on the brainstem surface could easily be identified and used to guide the level of sectioning. After sectioning, the preparation was pinned down with needles in a Sylgard-covered chamber mounted on a Zeiss Axioskop 2 microscope equipped with fluorescent filters and placed on a sliding table. The cut surface of the brainstem was held in the horizontal plane by crossing two pins under the rostral-most edge to lift and orient the cut surface to face the objective. We performed only one cut per preparation.

The cut surface of the brainstem $ChAT^{Cre};R26R^{TdTomato}$ preparation was illuminated with a fluorescent light source (Zeiss HXP120) for excitation (excitation filter, 550/25 nm) and visualization (emission filter, 605/70 nm), so the TdTom$^+$ motor nuclei could be visualized to verify the level of the cut. 5 N → 12 N equals to rostral → caudal level. Suction electrodes were placed on the lumbar left and right ventral roots to record locomotor activity in the lumbar spinal cord. Data were band-pass filtered (100 Hz to 1 kHz) and amplified 10,000-fold and sampled at 10 kHz with the pClamp software (Clampex v.10, Molecular

Devices). We then explored the entire transverse section for the effective sites where the electrical stimulation could evoke coordinated locomotor activity in the recorded ventral roots. Unilateral stimulation protocols using a glass pipette electrode with a tip diameter of ~70 μm was used. Trains of constant current pulses (5–20 Hz, 20–40 μA, 0.5 ms pulse duration) were delivered by an isolated unit driven by a Master-8 pulse generator (AMPI). The simulation was carried out continuously for 10–30 s.

A two-compartment system was used for pharmacological separation of the brainstem/upper spinal cord and lumbar spinal cord. For this, a Vaseline barrier was made at the level of the mid-thoracic spinal cord. Fast green dye was included in the rostral compartment to verify that the Vaseline barrier was not leaking, that is, drugs from the rostral compartment did not mix with the caudal compartment and vice versa. In several sets of experiments, kynurenic acid (KA, 4 mM; Sigma) was applied to the brainstem compartment to block glutamatergic transmission in the brainstem, thereby isolating the contribution of reticulospinal neurons to spinal circuits[24,39,40]. In other sets of experiments, KA or mephenesin (1 mM; Sigma) was applied in the caudal compartment to block or reduce polysynaptic transmission in the spinal cord[46].

### Optical identification of locomotor command regions

To test for the involvement of glutamatergic and/or serotonergic neurons in the locomotor command signal, in similar ways as electrical stimulation, we performed, an entire transverse cut for effective locomotor initiating sites using optical stimulation in crosses of $Vglut2^{Cre}$ and $SERT^{Cre}$ and $R26R^{ChR2-EYFP}$ mice. For stimulation, we used a 473 nm laser system (UGA-40; Rapp Optoelectronic), which delivered blue light at an intensity of 30 mW/mm$^2$ [105]. The blue light was directed at the preparation using an optical fiber (200 μm core, 0.22 NA, Thorlabs). The illumination was carried out continuously for 15–20 s. Two to four trials were tested for each preparation.

### Retrograde labeling

$Vglut2^{Cre}$ and $SERT^{Cre}$ mice were crossed with a conditional reporter line $R26R^{YFP}$. CTB conjugated with Alexa Fluor 555 (0.5% in saline, List Biological Laboratories) mixed with fast green dye was injected into the ventral spinal cord in anesthetized neonatal mice bilaterally in the L2 spinal segment In P0-P4 animals using a glass micropipette at a rate of 100 nL/min. The glass micropipette was held in place for 5 min following injection to prevent backflow, and the procedure was repeated on the other side of the cord. A total volume of 0.5 to 0.75 μL of CTB was injected on each side of the lumbar spinal cord. The skin was sutured after the injection. The injected spinal segment was verified by visualizing the fast green dye in the lumbar spinal cord during post-mortem dissection.

### Tissue immunochemistry, RNAscope®, imaging, and analysis

**Tissue immunochemistry.** Neonatal mice were euthanized by an overdose of isoflurane, 12 h after the CTB injection for the retrograde labeling study. The brain and spinal cord tissue were dissected free, and then post-fixed in 4% paraformaldehyde for 3 h at 4 °C. The tissue was cryoprotected by incubation in 30% sucrose in phosphate-buffered saline (PBS) overnight. The tissue was then embedded in Neg-50 medium (ThermoFisher Scientific) for cryostat sectioning. Transverse sections were obtained on a cryostat. Brainstem and spinal cord transverse sections were cut at 20–30 and 15–20 μm thickness, respectively. Sections were rehydrated for 5 min in PBS + 0.5% Triton-X100 (PBS-T; Sigma-Aldrich) and then blocked for 2 h in 10% normal donkey serum in PBS-T (Jackson ImmunoResearch). Sections were incubated overnight with primary antibodies diluted in a blocking solution. We used the following primary antibodies: chicken anti-GFP (1:1000, Abcam, ab13970), rabbit anti-tdTomato (1:1000, Takara, 632496), goat anti-CTB (1:1000, List Biological Laboratories #703),

mouse anti-TPH (1:2000, Millipore MAB5278) and rabbit anti-TPH2 (1:500, Millipore ABN60). Slides were washed four times in PBS-T and then incubated with appropriate fluorophore-coupled secondary antibodies diluted in blocking solution (all from Invitrogen; 1:500; Alexa-488 anti-chicken #A11039; Alexa-568 anti-rabbit #A10042; Alexa-555 anti-goat #A21432; Alexa-555 anti-mouse, #A21127). Slides were washed four times in PBS-T, counterstained with Hoechst 33342 (1:2000) or Nissl (NeuroTrace 435, 1:400, ThermoFisher Scientific), and were mounted with coverslips using Mowiol 4-88 medium. Sections were imaged using either a Zeiss widefield epifluorescence microscope, Zeiss LSM 700, or 900 confocal microscopes at the Core Facility for Integrated Microscopy, Copenhagen University. Anatomical imaging was collected with Zen (v3.5, Zeiss) and analyzed with ImageJ. Brainstem areas were delineated by systematically performing a Nissl/DAPI stain on sections. Coordinates and abbreviations are based on the neonatal mouse brain atlas by Paxinos et al.[106].

**RNAscope® in situ hybridization.** Neonatal mice were anesthetized with an overdose of isoflurane and sacrificed by decapitation. Brains were dissected and immediately snap frozen in isopentane kept on dry ice for cryoprotection. Coronal sections of the brainstem were cut (20 μm) and collected using a RNA free cryostat and Superfrost Plus slides (Thermo Fisher Scientific), respectively. All reagents used to perform the RNAscope® in situ hybridization were from the RNAscope® Multiplex Fluorescent Reagent Kit v2 (Advanced Cell Diagnostics (ACD) Bio-techne). A HybEZ™ oven was used for incubations, and samples were stored overnight at room temperature in 5× Saline Sodium Citrate after probe hybridization. The probes used were: *Vglut2* (slc17a6-C2, catalog number: 319171-C2, ACD Bio-techne) and *Sert* (Slc6a4-C1, catalog number: 315851, ACD Bio-techne). To visualize the probes, Opal™ 520 and Opal™ 570 fluorophores (1:1500, Akoya Biosciences) were used, respectively. Imaging was done with a confocal Zeiss 900.

**Analysis of locomotor activity evoked by brainstem stimulation**
Locomotor activity of the ventral roots evoked by brainstem stimulation was analyzed by the SpinalCore software[107]. The software uses Morlet wavelet algorithms on rectified data to derive the frequency component of the time series data recorded from each root. The time-frequency data are plotted with color-coded coherence values. The phase relationship between the signals is then presented as a vector field. Locomotor frequency and phase relationship were generated automatically by the software, in accordance with the preselected region of interest. Rayleigh's test was used to determine if the mean phase values reached statistical significance with a $p$ value <0.05.

**Ca²⁺ imaging recordings and analysis**
For visualizing activity in spinal interneurons of specific molecular identities, *Vglut2^Cre* and *Vgat^Cre* mice were crossed with a conditional reporter line (*R26R^GCaMP6f*), respectively. The spinal cord was transversally cut at the level of L3/4 and placed in a chamber mounted on the microscope with the transverse section facing the objective (×10, Fig. 3a). The recording field covers the entire transverse section. The cord was illuminated with the fluorescent light source for excitation (excitation filter, 470–490 nm) and visualization (emission filter, 520–560 nm). Activity-dependent changes in fluorescence were detected using a digital CMOS camera (PCO edge 5.5, Germany or Orca 4, Hamamatsu) at 10 frames/s and stored directly on the computer. Calcium data was collected with Camware (PCO) or HCImage (Hamamatsu). Changes in fluorescence were extracted offline using the image processing software, ImageJ. We, first, normalized the imaged recording field, where the four edges of the spinal cord meet the border of the field with the central canal at the ML center, to 500 × 400 pixels. Then, changes in fluorescence intensity over time for the normalized field were converted to $\Delta F = F_t - F_0$ where $F_t$ is the fluorescence at any specific time $t$ and $F_0$ is the baseline fluorescence (averaged

value of 100 frames prior to the stimulation onset). We generated 500 grid ROIs that cover the entire normalized field. $\Delta F$ variations over time were extracted for each ROI, normalized by the maximum $\Delta F$ value of all ROIs, and then reconstructed into a map for the entire transverse section. For analyzing the oscillation strength of the rhythmic Ca²⁺ activity for each of the ROIs, we low pass filtered and adjusted the baseline of the Ca²⁺ traces, and then carried out autocorrelation analysis by pClamp software (Clampfit v10.7, Molecular Devices), to evaluate the degree of regularity of rhythmicity. An autocorrelation of +1 represents a perfect positive correlation, while an autocorrelation of negative 1 represents a perfect negative correlation. The oscillation index was calculated as the value of the peak to the trough correlation coefficient. This value reflects the consistency and strength of the oscillatory activity. The larger the oscillation index, the stronger the rhythm is. A minimum value of 0.3 for individual ROIs was required to be included in further analyses[108]. Maps for oscillation index were generated for individual preparations, and then an averaged map is generated for qualifying group data. The ROI with the highest oscillation index of the averaged map was defined as the reference ROI, which was used in cross-correlation analysis (Clampfit v10.7, Molecular Devices). Cross-correlation between the individual ROIs and the reference ROI oscillation was used to reveal the wave traveling pattern between individual ROIs and the reference ROI. The phase value of individual ROIs was calculated, and the value was presented by the locomotor cycle duration. Phase maps were generated for individual preparations and then averaged for quantification purposes.

### Statistics
Statistical tests were performed using MATLAB (R2018b) or GraphPad (v.9, GraphPad Software). Normal distribution and homogeneity of variances were met, and repeated-measures analysis of variance (ANOVA) or paired $t$ test was used. Data are generally represented as box-whisker plots with medians, interquartile ranges were plotted (whiskers indicate minimal and maximal value). The $N$ values represent distinct biological replicates (mice), whereas the $n$ number indicates the trial or section number analyzed. Data in the text are presented as mean ± SD. The statistic significant threshold level was defined as $p < 0.05$.

### Reporting summary
Further information on research design is available in the Nature Portfolio Reporting Summary linked to this article.

## Data availability
Raw data files (i.e., videos and images) are not permanently deposited in an open access depository but are available from the corresponding author upon request. Source data are provided with this paper.

## Code availability
The code for analysis of calcium data can be made available upon request.

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

## Acknowledgements

We thank Dr. Martin Lauritzen for the $R26R^{GCaMP6f}$ mice and Dr. Aharon Lev-Tov and Dr. Yoav Mor for providing the SpinalCore software. We acknowledge the Core Facility for Integrated Microscopy, Faculty of Health and Medical Sciences, University of Copenhagen for using their microscopes. We thank Dr. Carmelo Bellardita for assistance with the calcium imaging, Dr. Jared Cregg for assistance with tracing, Iryna Vesth-Hansen for genotyping, Dr. Ilary Allodi and Roser Montañana-Rosell for assistance with the RNAscope® in situ hybridization experiments, Dr. Peter Löw for assistance with mice work, and members of Ole Kiehn's laboratory for discussion and comments on previous versions of this manuscript. This work was supported by the Lundbeck Foundation (postdoc grant L.-J.H. R219-2016-551), Swedish Research Council (O.K.), KID grant from Karolinska Institutet to M.B., the Novo Nordisk Foundation Laureate Program (NNF15OC0014186), and the Lundbeck Foundation (R276-2018-183 and R345-2020-1769).

## Author contributions

L.-J.H. and O.K. conceived the study. L.-J.H. and M.B. performed anatomical investigations, tracings, in vitro recordings, and analyzed data with contribution from O.K. L.-J.H. and O.K. wrote the paper with contribution from M.B. O.K. supervised all aspects of the work and provided the funding and technical support of the work.

## Competing interests

The authors declare no competing interests.
