## [Peer Review File · Nature Communications]

Deconstructing the modular organization and real-time dynamics of mammalian spinal locomotor networksREVIEWER COMMENTS

Reviewer #1 (Remarks to the Author):

The paper by Hsu, Bertho and Kiehn uses population Calcium imaging of the L3-L4 lumbar segment to describe the locomotor activity induced by brainstem stimulation.

I find the results interesting and important. In particular, the identification of a very localized brainstem region that can evoke locomotion is very impressive and it will certainly prompt further studies aimed at identifying the neurons involved. The Calcium imaging experiments extend throughout the spinal cord section, but of course the price to pay is the loss of cellular resolution. These days, 2-photon Calcium imaging of an entire spinal section has become technically possible, but certainly not mainstream enough to be implemented in any lab, so while one would dream of seeing these very same experiments performed at cellular resolution, the authors do a great job at extracting all the information from the technique they employ.

I really liked the paper and my only comments are requests for more information, because I am of the opinion that some of the data could have been reported better. Also, the methods section lacks some simple, but important detail, that I would ask the authors to add

Main comments:

Lines 151-159: I see the logic for performing CtB injections, but a few more details would be needed. It says three were n=11 CTB+ neurons, from N=3 animals, I suppose (2d). Then the same numbers n=11, N=3 are reported for CTB+/VGlut2+ neurons (2e) and again n=11, N=3 for CTB+/TPH+ (2f) neurons. I suggest to look again at the numbers, provide a breakdown of the number of CTB+, VGlut2+ and TPH+ neurons in each of the 3 animals. Also, Vglut2+ and TPH+ should also be expressed in relation to the total number of CTB+ neurons, as I assume that not all CTB+ cells are also Vglut2 (or TPH)

Line 198: It is stated here (and in methods) that the spinal cord imaging was done at lumbar segment 3/4. I understand the logic behind this choice, to include the most 'rhythmogenic' region (L1-2). However, I am wondering whether the fine temporal and spatial sequence of activation showed in figures 4-6 for glutamatergic and figure 7 for inhibitory neurons, is conserved across segments. In particular, I understand that imaging above L3-L4 would risk damaging the CPG, but I think it would be important to know if a similar observation (both the localization of the signal of figure 4 and the sequence of activation of figure 6) could be done at the L5 level. The imaged region is a 'neither flexor/nor extensor region in terms of motoneurons at least. Even though we do not know much about the rostrocaudal organization of the CPG, observing a similar hierarchy of activation also at the L5 level would be important. Not observing it, would be even more important and would highlight the need of looking also closer to L2.

Lines 260-261. It is stated here that the iEM remained active throughout the stimulation period. Was this activity rhythmic or tonic? From what I see in figure 3e, activity seems rhythmic. In general, I

understand the choice of format of figure 4b,d,f, but I think that a format like that of figure 3e would be more informative

Lines 275-282. It is stated here that the signal from motoneurons in ChAT-GCaMP mice is more ventrally located and smaller than the 'prM' area in the VGlut2-GCaMP experiments. Comparing Suppl. Figure 5h and figure 4m, I do not see the basis for this statement. I think this would be very important to quantify, because the iEM region is activated first and its area extends very close to what is defined as prM region, that in turn, is heavily overlapping with the motor nuclei. Whether the signals shown in the summary of Fig 4m originate at least in part from motoneurons, has some implication for the interpretation of the results and I think this point should be made crystal clear. One way would be to provide a proper quantification of the activated areas comparing the ChAT and Vglut2 experiments, but what I think is really needed is a proof of whether the Calcium signal can also come from motoneurons, because whether motoneurons express Vglut2 or not, is still very controversial, to put it mildly. The simplest way would be to image the VGlut2-GCaMP mice during ventral root stimulation. If there is a signal, then the area of the signal should be compared to that defined as prM during locomotion, while if there is no signal, the definition of prM as the set of premotor Interneurons activated during fictive locomotion would come out unambiguously. I think this point should be clarified: given the uncertainty in the geometric definition of the prM area, a convincing proof that it extends beyond the motor nuclei is needed

Methods: I find that there are a few procedures that are not fully explained in the methods. I list below the parts that I find lacking, but these will just require adding a few more details.

- 1) Spinal cord and brainstem sectioning: how were the cut performed? Vibratome? Blade? Also, when moving through more caudal brainstem section, was the sequential cuts performed on the same preparation?
- 2) Could the authors give a little more details on how the cu sections (both brainstem and spinal cord) were held in a horizontal plane in order to allow visualization and imaging of the tissue?
- 3) CTB injection: was the injection performed between vertebrae (without laminectomy)? Do the authors have an estimate of the site of injection in the L2 segment? Was it ventral or dorsal?
- 4) Line 592, tip diameter was 70 micron, not mm
- 5) Line 616: CTB conjugated with? Biotin? Alexa?
- 6) Please define oscillatory index: is it the peak to trough distance in the correlation coefficient? Also, please define the exact formula of the auto-correlation coefficient, because from plot 5c it looks like it is defined between [-1,1]

Minor

Line 112 Should be 'corresponds to' rather than 'corresponding to'

Line 169 'serotonergic neurons do not initiate locomotion' and I would add 'nor modulate it', as shown in Suppl. 3b, that is quite interesting (and to me, a bit unexpected)

Reviewer #2 (Remarks to the Author):

General Comments:

This manuscript describes an interesting and elegant description of the spatiotemporal organization of spinal modules involved in the transformation of tonic drive from descending regions of the brainstem into rhythmic activity in the spinal locomotor network. Overall, the work was received with enthusiasm, and it provides a roadmap for future work. Some weaknesses were identified, including a better rationale for unilateral stimulation and lack of control experiments related to electrical microstimulation, that, if addressed, will strengthen the manuscript.

Major Comments:

- The rationale for using unilateral stimulation is lacking and needs to be better articulated.
- Line 8: The use of the term “whole brain screening” is misleading and implies that more areas were examined than what was reported. A more accurate statement is “whole brainstem screening”.
- Lines 70-71: There is a lack of control experiments to support the authors assertion that lower intensity microelectrical stimulation (mES) restricts activation of smaller regions of the brainstem than does broader, more intense stimulation. Although this assertion is reasonable and likely, control experiments supporting this are suggested. Simply, one could express GCaMP6f in a pan-neuronal transgenic line and monitor/measure the spread of calcium signals with different stimulation parameters in the presence of TTX to abolish fast synapses. One predicts that the lower intensity mES will activate and spread across a smaller/more restricted region of the transverse cord. For example, mES of serotonergic region of transverse brainstem produced rhythmic spinal locomotor activity (Fig. 2h), which suggests that mES is not as restrictive as implied. Thus, there is not sufficient evidence to support the statement that mES is confined to the LPGi/CVL region (lines 179-82).

Minor Comments:

- Title: Grammatically incorrect and suggest either “Deconstruction of the modular...” or “Deconstructing the modular...”.
- Lines 122-23: The callouts to Fig. 1n for the left and right panels are reversed (left – time-frequency plot; right – circular plot).

- Lines 156-57 and Fig. 2: Panel e shows (labeled) colocalization of TPH and CTB, while panel f shows (labeled as) colocalization of Vglut2 and CBT, so it is necessary to confirm which subpanel goes with which callout.
- Lines 275-77: Please edit the sentence to reflect that calcium signal was monitored in spinal neurons but not in the ventral roots (as it is written). Extracellular recordings monitored activity in ventral roots.
- Line 336: Suggest to callout to Figure 3e regarding IEMi being the first rhythmic module.
- Line 345: LPGi should read LPGi.
- Lines 345-47: Provide the number of preparations that showed calcium signal in the ipsilateral lateral funiculus in response to LPGi/CVL stimulation in KA and mephenesin.
- Lines 347-49: Provide the number of preparations that showed calcium signal in neurons close to the central canal in response to LPGi/CVL stimulation in mephenesin.
- Figures 1 and Supplementary Figure 1: The sagittal cuts located at 12N in Fig. 1 are listed as “12NR” and “12NC”, whereas the same cuts in Supp. Fig. 1 are listed as “12-1” and “12N-2”. So, the reader must assume which cuts are analogous in the two figures... please clarify.

Reviewer #3 (Remarks to the Author):

In this manuscript, Hsu et al describe their work outlining a descending locomotor command from a specific brain stem reticulospinal area, the CVL/LPGi, to the mid-lumbar spinal cord, where the tonic descending signal is converted to a rhythmic signal to produce locomotor-like activity. They are addressing an important question – one that will be of great interest to the community studying locomotion. In addition, they use a broad range of techniques, and demonstrate the use of those techniques to understand connectivity between distant regions of the nervous system. Furthermore, the study is well organised and logical, and beautifully illustrated. As such, I am very enthusiastic about it.

Nonetheless, I have a few major concerns. I hope the authors read this in light of my enthusiasm and don't get discouraged by the length or number of my comments!

The manuscript is looking at the initiation of locomotion in the neonatal mouse over space and time. There are issues about both that the authors should consider addressing:

1. Space: The authors are looking at a single segment in the lumbar spinal cord, L3/4. It is not clear why they selected this segment, and how the results would have differed, and the implications of those differences, had they looked elsewhere. For example, in the senior author's earlier work in the neonatal rat, the suggestion was made that the key region was in the more rostral lumbar spinal cord. While the authors need not repeat experiments at different segments (although this would be nice!), there should at least be some thoughtful discussion about this shortcoming.

2. Time: The authors use a 10 fps camera, limiting their imaging to 100 ms frames. Of course, events are much faster than this, so the "averaging" over 100 ms may diminish or even miss some events. It would be good to know where the first synaptic events occur. Again, while it would be nice to have a look at a higher frame rate, I think that more fully discussing this shortcoming would be sufficient.

Other major concerns:

3. Previous studies of relevance: The main study that is missing is one from the Jordan lab, where that group used field potential mapping in the cat spinal cord to study where descending locomotor signals arrived. I really think this is of major importance to the current manuscript, and that the results should be compared. The Jordan group used a technique with much faster time resolution (electrophysiology) than calcium imaging, and explored 4 segments in the cat spinal cord. (See PMID: 7891162)

There are other relevant cat studies relegated to the history books, many of which are cited here. But the authors state that previous studies have only stimulated "broader areas of the MRF" (lines 382-383). This is not accurate, as microstimulation studies in the cat have nicely demonstrated the Gi as being a locomotor-initiating site. Perhaps the senior author remembers the amazing videos shown by Shigemori of such studies in intact cats presented at the Wenner Gren meeting in 1985? (See PMID: 2611678.)

This also brings up another point: there seem to be some interspecies differences, given the more medial sites important in the cat (see also Noga's work) and the more lateral regions (CVL/LPGi) in the mouse. It would be good to have some discussion about this issue. (Interestingly, note also that the "several seconds" to initiate locomotion in the mouse (lines 479-480) is also the case in the cat.)

And this leads me to the issue of the Pontine Locomotor Strip, which it seems the authors have identified in the mouse. Early work in the cat from several labs (I can recall Jordan, Garcia-Rill, and maybe Mori as well as Shik, Orlovskii et al) supported that this was a polysynaptic pathway, with distributed synapses including in the cervical cord (Russian group in my memory, but it was a long time ago). The PLS was thought to be located near nV/Vsp, so it fits with the work presented here. But I don't

think it's necessary to say that it functions through the LPGi/CVL (lines 402-403), given the cat data (which the authors may want to discuss a bit more?).

4. Stimulation sites. In addition to the above comment about CVL/LPGi vs Gi and possible species differences, I am struck by the result (line 88-89) that the CVL is a stronger site than the LPGi. Two questions here: (a) how does that fit with the data from the Arber lab, which points to the LPGi? And (b) The CVL is closely associated with the pre-Bötzing complex, a key respiratory centre... given relationships between respiration and locomotion, is this relevant and is it something that should be discussed? (E.g. could this centre be to coordinate the two, such as ensuring that expiration occurs at the biomechanically advantageous point of foot fall?)

5. The explanations sometimes go beyond the data. There is really no reason to over-interpret cause and effect, as the data are excellent as they are. In their concluding remarks, the authors hit the nail on the head (and the tone), and this view should be expressed throughout the manuscript rather than trying to make too much of the data. Specific examples include:

☐ Line 133, “the final descending command signal” – I would agree that this is “a” descending command signal for locomotion, but to say it’s “the” pathway means that there are no others (e.g. what about multisynaptic pathways, see PLS point above?).

☐ The descending command leads to activity in excitatory and inhibitory neurons, and their location is well defined in this study. Furthermore, the argument is nicely made that the synapses with glutamatergic neurons are key. But to say that the descending command is “executed by” (line 232) these modules goes a bit beyond... maybe the descending command must go to both excitatory and inhibitory neurons for locomotion to be effected, for example?

☐ Lines 234-235 – first part, facilitating rhythm generation, okay. But how do we know that the “modules” work together?

☐ Lines 377-378 are really too much. The “how” is not addressed here. To understand how it's transformed, we need to understand the specific cells that it's activating, the intrinsic properties of those cells, and their integration with specific circuits used to generate the rhythm. Same as the “how” in lines 490 and again in 491 (which should be “where” not “how”).

☐ Line 460 – “specific function” – this is beyond what the experiments tested.

☐ Line 485 – “amplifies the oscillations” – I didn't see this tested anywhere in the manuscript.

6. I have been oscillating like the figures on the contribution of the vGAT-GCaMP experiments, and have plateaued at the point where I think they are useful and interesting. But I think they are superficially described and illustrated, and conclude that this section should be strengthened. The vGluT2 data are so well illustrated. For example, Fig 1e is really nice: the section/figure on inhibitory neurons would be well served by a figure like this.

7. Locomotion: given that the cord was cut at L3/4, it's hard to define the rhythmic activity as "locomotion" per se, in which you need alternating activity in flexor and extensor motor neurons. That is, while I think it's okay to use the term most of the time as the evidence is good, I think this caveat should be expressed at the outset.

8. Standard error of the mean: This is a completely useless number and should be eliminated from all figures. In biology, we're interested in variability, not where a "true" mean is. As such, at minimum, the standard deviation should be shown. Better would be box-whisker plots with medians, interquartile ranges, etc.

9. Modules: I think it's okay to use this word. I'll just point out that I think to many (or at least to me), the word refers to particular microcircuits (see, e.g., El Manira zebrafish work), which weren't studied here (i.e. not at the neuronal level). So while I would go for the word "regions" here, it's up to the authors.

Minor:

One general, minor comment that does not need a response. The 5HT work was very interesting. While I think most investigators would say that the descending command is glutamatergic (as the authors cite), many would say that 5HT plays an important auxiliary role. As such, Suppl fig 3 is very interesting in that there does not seem to be a modulating effect. It would be interesting, one day, to stimulate LPGi and add 5HT blockers!

1. Title: Grammatically incorrect, and should read "Deconstructing" for example.

2. Introduction, 1st paragraph. The term "executive" is incorrect. When it comes to the CNS, executive function involves higher cognitive thought, and I don't think the authors mean that's what the spinal cord is doing. I think they mean, perhaps, "executing." Also in line 154.

3. Line 42, remove "that"

4. Line 102 – locomotor efficiency, and locomotor index in the figure – I cannot find how efficiency was defined, or how index was calculated.

5. Line 179, perhaps defined instead of confined?

6. Figure 2n is not clear, and the legend doesn't help much. Could be improved.

7. Perhaps it's simply my lack of expertise, but the authors are suggesting repeatedly that calcium signal can be seen in descending axons. How does this work? How is the calcium entering? I suppose that

these RSN axons have calcium channels (what kind, for what?) at their nodes of Ranvier, or are perhaps still unmyelinated at this age? It might be nice to add a couple of statements about this somewhere. Has a similar finding of Ca activity in long axons been previously described?

8. Line 220, “slower onset” should be quantified

9. Line 259: when reading this, one wonders why it takes 100 ms. Much later, it becomes clear that that’s the time resolution of the experiments. Best to mention that here so the reader can interpret.

10. Lines 349-350 – just a comment about some preps vs others – this could reflect the clustering of these neurons in the rostro-caudal dimension, and sometimes you don’t hit the clusters? No response needed.

11. Line 353 – not clear how the authors can say definitively that these are “non locomotor initiating”?

12. Lines 389-390: “without having to penetrate tissue” is a false argument. Penetration does little harm, and certainly much less than lopping off rostral regions!

13. Line 395: “impossible” is a pretty strong word

14. Line 438, maybe concurrently instead of coherently?

Methods:

1. Dissection – temperature of recordings should be indicated

2. Lines 585-586, verifying level of the cut – this isn’t clear at all. How?

3. The nonparametric tests were not defined. When and how often were the data normally distributed?

Response to referees

We would like to thank all reviewers for their insightful and positive comments about our work. We have performed a thorough revision of our manuscript and added new data analysis and experiments to support our conclusion as outlined in detail below.

Reviewer #1 (Remarks to the Author):

The paper by Hsu, Bertho and Kiehn uses population Calcium imaging of the L3-L4 lumbar segment to describe the locomotor activity induced by brainstem stimulation. I find the results interesting and important. In particular, the identification of a very localized brainstem region that can evoke locomotion is very impressive and it will certainly prompt further studies aimed at identifying the neurons involved. The Calcium imaging experiments extend throughout the spinal cord section, but of course the price to pay is the loss of cellular resolution. These days, 2-photon Calcium imaging of an entire spinal section has become technically possible, but certainly not mainstream enough to be implemented in any lab, so while one would dream of seeing these very same experiments performed at cellular resolution, the authors do a great job at extracting all the information from the technique they employ.

I really liked the paper and my only comments are requests for more information, because I am of the opinion that some of the data could have been reported better. Also, the methods section lacks some simple, but important detail, that I would ask the authors to add.

We are happy that the reviewer likes our manuscript. We have added quantification of anatomy data and performed new experiments to clarify the issue about glutamate calcium signal in motor neurons as outlined below. We have also thoroughly discussed our choices of recorded lumbar segment.

Main comments:

Lines 151-159: I see the logic for performing CtB injections, but a few more details would be needed. It says three were $n=11$ CTB+ neurons, from $N=3$ animals, I suppose (2d). Then the same numbers $n=11$, $N=3$ are reported for CTB+/VGlut2+ neurons (2e) and again $n=11$, $N=3$ for CTB+/TPH+ (2f) neurons. I suggest to look again at the numbers, provide a breakdown of the number of CTB+, VGlut2+ and TPH+ neurons in each of the 3 animals. Also, Vglut2+ and TPH+ should also be expressed in relation to the total number of CTB+ neurons, as I assume that not all CTB+ cells are also Vglut2 (or TPH)

The reviewer is absolutely correct. We did not provide these data in a comprehensive way. We have added new data with RNAscope® *in situ* hybridization to show the expression of Vglut2 and serotonergic cells in the region. These data show nicely the

localization of Vglut2⁺ and Sert⁺ (serotonergic) cells in LPGi/CVL compared to our previous *in situ* hybridization with immunocytochemistry data (Page 6, Line 151-153). Next, we have now reported the CTB data as suggested by the reviewer with clear data from 2 animals instead of 3. Because of the relatively faint YFP expression, we expect that there should be a higher co-localization of Vglut2 and CTB cells than we are able to detect. However, the conclusion remain the same: glutamatergic and serotonergic reticulospinal neurons are located in the LPGi/CVL area (Page 6, Line 165-171).

Line 198: It is stated here (and in methods) that the spinal cord imaging was done at lumbar segment 3/4. I understand the logic behind this choice, to include the most 'rhythmogenic' region (L1-2). However, I am wondering whether the fine temporal and spatial sequence of activation showed in figures 4-6 for glutamatergic and figure 7 for inhibitory neurons, is conserved across segments. In particular, I understand that imaging above L3-L4 would risk damaging the CPG, but I think it would be important to know if a similar observation (both the localization of the signal of figure 4 and the sequence of activation of figure 6) could be done at the L5 level. The imaged region is a 'neither flexor/nor extensor region in terms of motoneurons at least. Even though we do not know much about the rostrocaudal organization of the CPG, observing a similar hierarchy of activation also at the L5 level would be important. Not observing it, would be even more important and would highlight the need of looking also closer to L2.

We understand the point about conservation of the pattern throughout the extend of the lumbar spinal cord (L1-L6) that contains the elements of CPG for hindlimb locomotion. This is also a point that is raised by reviewer 3 who want us to discuss this issue but not performing further experiments. We were careful in our choice of the segments because from previous experiments we know (and this is well known in the field) that if we damage L1-L2 the rhythmic activity tends to disappear. But we also know that L3 and caudally located segments can generate a rhythm in isolation form L1-L2 (see Kjaerulff and Kiehn 1998; Kjaerulff and Kiehn 1996; Hagglund et al. 2013). We therefore choose 'L3-L4' in close proximity to L1-L2. We did not systematically investigate the hierarchy of activation at the level of L5. We do agree with the reviewer that it would be interesting to have such experiments in the future. However, these experiments are cumbersome and will take a long time to perform, and we are not sure that they are needed for this study (See also comments from reviewer 3). We, therefore, have not done those experiments. In acknowledgment of the issue we have, however, now added a discussion (Page 20-21, Line 526-540) of the distribution in the lumbar region with also includes a comparison with data from the brilliant Jordan paper 1995 that studied the effect of a descending signal from MLR stimulation in the cat spinal cord.

Lines 260-261. It is stated here that the iEM remained active throughout the stimulation period. Was this activity rhythmic or tonic? From what I see in figure 3e,

activity seems rhythmic. In general, I understand the choice of format of figure 4b,d,f, but I think that a format like that of figure 3e would be more informative.

We might not have been clear enough about this point. But iEM is first tonically active in the initiation phase and then switch immediately to rhythmicity in the sustained rhythmic phase (see Fig. 3e and g). We have maintained the representation in Fig. 4.

Lines 275-282. It is stated here that the signal from motoneurons in ChAT-GCaMP mice is more ventrally located and smaller than the 'prM' area in the VGlut2-GCaMP experiments. Comparing Suppl. Figure 5h and figure 4m, I do not see the basis for this statement. I think this would be very important to quantify, because the iEM region is activated first and its area extends very close to what is defined as prM region, that in turn, is heavily overlapping with the motor nuclei. Whether the signals shown in the summary of Fig 4m originate at least in part from motoneurons, has some implication for the interpretation of the results and I think this point should be made crystal clear. One way would be to provide a proper quantification of the activated areas comparing the ChAT and Vglut2 experiments, but what I think is really needed is a proof of whether the Calcium signal can also come from motoneurons, because whether motoneurons express Vglut2 or not, is still very controversial, to put it mildly. The simplest way would be to image the VGlut2-GCaMP mice during ventral root stimulation. If there is a signal, then the area of the signal should be compared to that defined as prM during locomotion, while if there is no signal, the definition of prM as the set of premotor Interneurons activated during fictive locomotion would come out unambiguously. I think this point should be clarified: given the uncertainty in the geometric definition of the prM area, a convincing proof that it extends beyond the motor nuclei is needed.

The reviewer is right in the view about the distinction between pMN and motor neurons. We are aware of this issue. *We now have done new experiments with antidromic activation of motor neurons and calcium imaging to address the issue as suggested by the reviewer.* We thank the reviewer for this suggestion. These new experiments show that motor neurons contribute very little to the calcium signal in the Vglut2GCaMP6f mouse. This finding makes the definition of prM as a set of premotor interneurons very clear and thus strengthens our original conclusion about prM. **These new experiments are shown in Supplementary Fig. 8** and the text reads as below. Page 12, starting line 288:

"The close proximity of the premotor module to the motor neuron pool and the fact that Vglut2 has been shown to be expressed weakly in motor neurons⁵⁰ raise the possibility that at least part of the signal in the prM module could be from motor neurons. To further qualify this conjecture, we performed Ca²⁺ imaging in Vglut2^{Cre}; R26R^{GCaMP6f} mice while we antidromically activated motor neurons by ventral root stimulation (4 Hz, pulse duration 200 μs, 10 s, 120-300 μA) on one side of the cord with simultaneous recording of the ventral root activity on the other side of the cord at the same segmental level. Under aCSF perfusion, ventral root stimulation might evoke bilateral rhythmic activity⁵¹ because of release of glutamate from central motor neurons collaterals^{50, 51}. As an indication that the ventral root

*stimulation indeed activates the motor neurons on the stimulated side, we found activity in the contralateral ventral root which sometimes was rhythmic. A clear Ca^{2+} signal was present in the intermediate area around the central canal with a weaker signal close to or in the motor neuron pools (Supplementary Fig. 8 a-d, N=3). When nicotinic receptors and glutamatergic receptors were blocked – thereby isolating the antidromic stimulation to the motor neuron pool (by removing any effect of central motor neuron collaterals on spinal circuits) - there was no calcium signal in intermediate area and only a very weak calcium signal in the stimulated motor neuron pool (Supplementary Fig. 8 e-i, N=3). These data demonstrate that while motor neurons indeed give a weak ventrally located calcium signal in *Vglut2^{Cre}; R26^{RCaMP6f}* – the contribution of this signal is minor and negligible to what we call the premotor module (prM). We also performed calcium imaging in *ChAT^{Cre}; R26^{RCaMP6f}* mice. ”*

Methods: I find that there are a few procedures that are not fully explained in the methods. I list below the parts that I find lacking, but these will just require adding a few more details.

1) Spinal cord and brainstem sectioning: how were the cut performed? Vibratome? Blade? Also, when moving through more caudal brainstem section, was the sequential cuts performed on the same preparation?

We have provided a better explanation of the methods (Page 23, Line 635, 641) to answer these questions.

2) Could the authors give a little more details on how the cu sections (both brainstem and spinal cord) were held in a horizontal plane in order to allow visualization and imaging of the tissue?

Explanation is now added on Page 23, Line 639-641.

3) CTB injection: was the injection performed between vertebrae (without laminectomy)? Do the authors have an estimate of the site of injection in the L2 segment? Was it ventral or dorsal?

Explanation is now added on Page 24, Line 676-677 ; Line 682-683.

4) Line 592, tip diameter was 70 micron, not mm.

Thanks for pointing it out. Texts are now changed (Page 23, Line 651).

5) Line 616: CTB conjugated with? Biotin? Alexa?

Alexa Fluor 555. This information is now mentioned (Page 24, Line 676).

6) Please define oscillatory index: is it the peak to trough distance in the correlation coefficient? Also, please define the exact formula of the auto-correlation coefficient, because from plot 5c it looks like it is defined between [-1,1]

Explanation was given in the Method and is now expanded on Page 25, Line 750-754. The formula is from a commercially available software – Clampfit - and is standard.

Minor

Line 112 Should be 'corresponds to' rather than 'corresponding to'
Text is now changed (Page 5, Line 117).

Line 169 'serotonergic neurons do not initiate locomotion' and I would add 'nor modulate it', as shown in Suppl. 3b, that is quite interesting (and to me, a bit unexpected)
Changed as requested (Page 7, Line 183-184).

Reviewer #2 (Remarks to the Author):

General Comments:

This manuscript describes an interesting and elegant description of the spatiotemporal organization of spinal modules involved in the transformation of tonic drive from descending regions of the brainstem into rhythmic activity in the spinal locomotor network. Overall, the work was received with enthusiasm, and it provides a roadmap for future work. Some weaknesses were identified, including a better rationale for unilateral stimulation and lack of control experiments related to electrical microstimulation, that, if addressed, will strengthen the manuscript.

Thanks for the positive stroke. We have addressed the issues raised by clarifying the text and additional analysis.

Major Comments:

- The rationale for using unilateral stimulation is lacking and needs to be better articulated.

Thanks for this comment. We envisaged that it was more straightforward and readily understandable to use unilateral stimulation and mapping as it has also been done traditionally in brainstem mapping experiments. With the active site in hand it also turns out that unilateral (as has also been known for years of MLR stimulation) induce bilateral locomotion. Moreover, we find that the unilateral stimulation leads to a slight asymmetric activation of the spinal locomotor networks which was advantageous for our analysis since we could follow the signal travelling from the ipsilateral side in the cord to the contralateral. We have clarified our rationale on Page 3, Line 86-88.

- Line 8: The use of the term "whole brain screening" is misleading and implies that

more areas were examined than what was reported. A more accurate statement is “whole brainstem screening”.

The reviewer is right. It should not be whole brain screening but “whole brainstem screening”. Text is now changed (Page 1, Line 8).

- Lines 70-71: There is a lack of control experiments to support the authors assertion that lower intensity microelectrical stimulation (mES) restricts activation of smaller regions of the brainstem than does broader, more intense stimulation. Although this assertion is reasonable and likely, control experiments supporting this are suggested. Simply, one could express GCaMP6f in a pan-neuronal transgenic line and monitor/measure the spread of calcium signals with different stimulation parameters in the presence of TTX to abolish fast synapses. One predicts that the lower intensity mES will activate and spread across a smaller/more restricted region of the transverse cord. For example, mES of serotonergic region of transverse brainstem produced rhythmic spinal locomotor activity (Fig. 2h), which suggests that mES is not as restrictive as implied (this is a misunderstanding). Thus, there is not sufficient evidence to support the statement that mES is confined to the LPGi/CVL region (lines 179-82).

When we referred to intense stimulation in previous locomotor experiments in the rodent: they used 1 mA which 10-50 times stronger than the stimulus strength we used. The minimal threshold strength we use is 20-30 μ A similar to brainstem mapping experiments used by Glover and Perreault (2008, 2011, 2014). They estimated a current spread from the tip of electrode to be around 200-300 μ m at 40 μ A. The same calculation could apply to our experiments. Importantly, we carefully mapped the stimulation from the surface by visually placing the stimulation electrode in equidistant points in a matrix covering the entire surface of the cut brainstem. Close to (\sim 350 μ m) active points we found inactive points which shows the current did not spread to the active points. *Based on these arguments we retain that our stimulation protocol is reliable and with internal points inactive points as control.*

However, to further address this issue ***we have performed extra experiments as suggested by the reviewer*** using the Vglut2^{Cre};R26R^{GCaMP6f} mouse line and measuring the calcium change in the brainstem after blocking glutamatergic collaterals with KA. These experiments are presented in **Supplementary figure 1** along with text edits (Page 2-3, Line 71-77). They show that there is limited calcium activation (\sim 350 μ m) around the electrode at low current amplitudes. The spread increases with higher intensities. *We therefore retain that there is evidence to support the statement that mES is confined to the LPGi/CVL region.*

Minor Comments:

- Title: Grammatically incorrect and suggest either “Deconstruction of the modular...” or “Deconstructing the modular...”.

Thanks for pointing it out. Title is now changed.

- Lines 122-23: The callouts to Fig. 1n for the left and right panels are reversed (left – time-frequency plot; right – circular plot).

Thanks for pointing out. Texts are now changed (Page 5, Line 137-138).

- Lines 156-57 and Fig. 2: Panel e shows (labeled) colocalization of TPH and CTB, while panel f shows (labeled as) colocalization of Vglut2 and CBT, so it is necessary to confirm which subpanel goes with which callout.

Thanks for pointing out. Texts are now changed (Page 6, Line 168 and 170).

- Lines 275-77: Please edit the sentence to reflect that calcium signal was monitored in spinal neurons but not in the ventral roots (as it is written). Extracellular recordings monitored activity in ventral roots.

Text edited as requested (Page 13, Line 307-309).

- Line 336: Suggest to callout to Figure 3e regarding IEMi being the first rhythmic module.

Text edited as requested (Page 16, Line 369).

- Line 345: LPGi should read LPGi.

Thanks for pointing out. Texts are now changed (Page 16, Line 378).

- Lines 345-47: Provide the number of preparations that showed calcium signal in the ipsilateral lateral funiculus in response to LPGi/CVL stimulation in KA and mephenesin.

We now have provided the requested number (Page 16, Line 379-380).

- Lines 347-49: Provide the number of preparations that showed calcium signal in neurons close to the central canal in in response to LPGi/CVL stimulation in mephenesin.

We now have provided the requested number (Page 16, Line 380).

- Figures 1 and Supplementary Figure 1: The sagittal cuts located at 12N in Fig. 1 are listed as “12NR” and “12NC”, whereas the same cuts in Supp. Fig. 1 are listed as “12-1” and “12N-2”. So, the reader must assume which cuts are analogous in the two figures... please clarify.

Thank you for point this out. Texts are now clarified in the figures.

Reviewer #3 (Remarks to the Author):

In this manuscript, Hsu et al describe their work outlining a descending locomotor command from a specific brain stem reticulospinal area, the CVL/LPGi, to the mid-lumbar spinal cord, where the tonic descending signal is converted to a rhythmic signal to produce locomotor-like activity. They are addressing an important question – one that will be of great interest to the community studying locomotion. In addition, they use a broad range of techniques, and demonstrate the use of those techniques to understand connectivity between distant regions of the nervous system. Furthermore, the study is well organised and logical, and beautifully illustrated. As such, I am very enthusiastic about it.

Nonetheless, I have a few major concerns. I hope the authors read this in light of my enthusiasm and don't get discouraged by the length or number of my comments!

We would like to thank this reviewer for very insightful comments. The reviewer is right in most issues raised. We have as requested added discussion points and clarifications to all request as outlined in detail below.

The manuscript is looking at the initiation of locomotion in the neonatal mouse over space and time. There are issues about both that the authors should consider addressing:

1. Space: The authors are looking at a single segment in the lumbar spinal cord, L3/4. It is not clear why they selected this segment, and how the results would have differed, and the implications of those differences, had they looked elsewhere. For example, in the senior author's earlier work in the neonatal rat, the suggestion was made that the key region was in the more rostral lumbar spinal cord. While the authors **need not repeat** experiments at different segments (although this would be nice!), there should at least be some thoughtful discussion about this shortcoming. We understand the point about conservation of the pattern throughout the lumbar spinal cord (L1-L6) that contains the elements of CPG for hindlimb locomotion. This is also a point that is raised by reviewer 1. We were careful in our choice of the segments because from previous experiments we know (and this is well known in the field) that if we damage L1-L2, the rhythmic activity tends to disappear. But we also know that L3 and caudally located segments can generate a rhythm in isolation from L1-L2 (see Kjaerulff and Kiehn 1998; Kjaerulff and Kiehn 1996; Hagglund et al. 2013). We therefore choose 'L3-L4' in close proximity to L1-L2. We did not systematically study the hierarchy of activation at the level of L5. We do agree with the reviewer that it would be interesting to have such experiments in the future. However, these

experiments are cumbersome and will take a long time perform. We therefore have not done those experiments. In acknowledgment of the issue, we have now added a discussion of the distribution in the lumbar region with also includes a comparison with data from the brilliant Jordan paper 1985 that studied the effect of a descending signal from MLR stimulation in the cat spinal cord (Page 20-21, Line 526-540).

2. Time: The authors use a 10 fps camera, limiting their imaging to 100 ms frames. Of course, events are much faster than this, so the “averaging” over 100 ms may diminish or even miss some events. It would be good to know where the first synaptic events occur. Again, while it would be nice to have a look at a higher frame rate, I think that more fully discussing this shortcoming would be sufficient.

We already had some discussion of this issue in the current version of the manuscript where we clearly stated that we cannot determine monosynaptic latencies based on these frame rates (Page 20, Line 507-520) but that the pharmacology allow us to that

Other major concerns:

3. Previous studies of relevance: The main study that is missing is one from the Jordan lab, where that group used field potential mapping in the cat spinal cord to study where descending locomotor signals arrived. I really think this is of major importance to the current manuscript, and that the results should be compared. The Jordan group used a technique with much faster time resolution (electrophysiology) than calcium imaging, and explored 4 segments in the cat spinal cord. (See PMID: 7891162)

The reviewer is right. It is a mistake that we did not discuss and compare our data with this extremely nice study from Larry Jordan. We now give full credit to the cat study and discuss the similarities and differences to our study on Page 20-21, Line 526-540.

4. There are other relevant cat studies relegated to the history books, many of which are cited here. But the authors state that previous studies have only stimulated “broader areas of the MRF” (lines 382-383). This is not accurate, as microstimulation studies in the cat have nicely demonstrated the Gi as being a locomotor-initiating site. Perhaps the senior author remembers the amazing videos shown by Shigemi Mori of such studies in intact cats presented at the Wenner Gren meeting in 1985? (See PMID: 2611678.)

The senior author indeed remember Shigemi Mori’s nice studies presented at the Wenner Gren meeting in 1985. The data were later published in 1989 and indeed show micro-stimulation can evoke inhibitory and excitatory responses in MRF and other studies in both the cat and rodent (Perrault and Glover) have done that. We have deleted that sentence from the manuscript since it is misleading (Page 18, Line

409-411) and rephrase the sentence and added an appropriate discussion (Page 18, Line 436-441).

This also brings up another point: there seem to be some interspecies differences, given the more medial sites important in the cat (see also Noga's work) and the more lateral regions (CVL/LPGi) in the mouse. It would be good to have some discussion about this issue. (Interestingly, note also that the "several seconds" to initiate locomotion in the mouse (lines 479-480) is also the case in the cat.)

We recognize that the optimal locomotor site is in Gi in cats, and we now refer to those papers directly and discuss the differences to the mouse (Page 18, Line 436-441). We are quite aware that there are "several seconds" to initiate locomotion also in the cat and also give references to the manuscript (Page 20, Line 521-525).

And this leads me to the issue of the Pontine Locomotor Strip, which it seems the authors have identified in the mouse. Early work in the cat from several labs (I can recall Jordan, Garcia-Rill, and maybe Mori as well as Shik, Orlovskii et al) supported that this was a polysynaptic pathway, with distributed synapses including in the cervical cord (Russian group in my memory, but it was a long time ago). The PLS was thought to be located near nV/Vsp, so it fits with the work presented here. But I don't think it's necessary to say that it functions through the LPGi/CVL (lines 402-403), given the cat data (which the authors may want to discuss a bit more?).

We are happy that the referee recognizes this and agree with us – we indeed already referred to PLS in our previous version. We now add two new references to this work in the paper from Mori and Shik (1977 and 1978). The reviewer is right that there is a possibility that the PLS does not act through LPGi/CVL and we just block a polysynaptic pathway. This does not change our conclusion, but we have included this interpretation as well on Page 18, Line 424-432.

"Indeed, we show that non-cell-specific micro-electrical stimulation of the lateral medulla along the rostrocaudal levels evokes locomotor bursts, with a locomotor-like pattern. These areas form a strip, which corresponds to the previously identified pontomedullary locomotor strip, a descending tract that evokes locomotion in cats^{15, 17, 65-68}. However, the electrically induced locomotor-like bursts were absent or largely diminished after blocking the collateral glutamatergic activity in the brainstem and upper spinal cord, suggesting that these areas mediate their effect through collaterals in the brainstem, possibly acting through the LPGi/CVL or through other polysynaptic relays in brainstem or upper spinal cord."

4. Stimulation sites. In addition to the above comment about CVL/LPGi vs Gi and possible species differences, I am struck by the result (line 88-89) that the CVL is a stronger site than the LPGi. Two questions here: (a) how does that fit with the data from the Arber lab, which points to the LPGi?

We have changed the text to comment on this issue. Page 19, Line 453-459:

"Nevertheless, the convergence of LPGi in neonatal and adult mice suggests that LPGi is ontogenetically preserved for locomotor initiation. However, we have also found that the locomotor-initiating area includes CVL in the neonatal mouse. This apparent difference

between neonatal and adult mice may reflect developmental changes. Nevertheless, because of the intact glutamatergic transmission in the brainstem in adult mouse experiments it seems difficult to completely exclude a contribution from CVL even though it was not stimulated directly. "

And (b) The CVL is closely associated with the pre-Bötzinger complex, a key respiratory centre... given relationships between respiration and locomotion, is this relevant and is it something that should be discussed? (E.g. could this centre be to coordinate the two, such as ensuring that expiration occurs at the biomechanically advantageous point of foot fall?)

We are not experts on pre-Bötzinger complex. But we think that the pre-Bötzinger complex is located a bit more caudal than the CVL (Ruangkittisakul...Del Negro, 2014, PMID: 25138790) that we identify as a locomotor site. CVL is close to the parafacial respiratory group though, and we mention this in the text now (Page 18, Line 442-445).

5. The explanations sometimes go beyond the data.

Yes, we agree – we have toned down the text in the places pointed out by the reviewer.

There is really no reason to over-interpret cause and effect, as the data are excellent as they are. In their concluding remarks, the authors hit the nail on the head (and the tone), and this view should be expressed throughout the manuscript rather than trying to make too much of the data. Specific examples include:

♣ Line 133, "the final descending command signal" – I would agree that this is "a" descending command signal for locomotion, but to say it's "the" pathway means that there are no others (e.g. what about multisynaptic pathways, see PLS point above?). Changed to 'a' (Page 6, Line 141).

♣ The descending command leads to activity in excitatory and inhibitory neurons, and their location is well defined in this study. Furthermore, the argument is nicely made that the synapses with glutamatergic neurons are key. But to say that the descending command is "executed by" (line 232) these modules goes a bit beyond... maybe the descending command must go to both excitatory and inhibitory neurons for locomotion to be effected, for example?

We have toned down the text and now replace it with the following texts: (Page 10-11, Line 246-250).

"These results demonstrate that the unilateral LPGi/CVL command is received by excitatory spinal modules that leads to the expression of: first a tonic initiation phase during which excitatory modules in the spinal cord are recruited to facilitate rhythm generation, and then in a rhythmic phase in which the modules are active to drive the locomotor-like activity."

♣ Lines 234-235 – first part, facilitating rhythm generation, okay. But how do we know that the "modules" work together?

Agree. This is implied from the indirect observation that they are active in different part of the cycle. We have changed the wording slightly (Page 11, Line 250).

♣ Lines 377-378 are really too much. The “how” is not addressed here. To understand how it's transformed, we need to understand the specific cells that it's activating, the intrinsic properties of those cells, and their integration with specific circuits used to generate the rhythm. Same as the “how” in lines 490 and again in 491 (which should be “where” not “how”).

We agree- how is changed to when and where (Page 17, Line 405-406).

♣ Line 460 – “specific function” – this is beyond what the experiments tested.

Deleted (Page 20, Line 503).

♣ Line 485 – “amplifies the oscillations” – I didn't see this tested anywhere in the manuscript.

Agree – wording deleted (Page 21, Line 544).

6. I have been oscillating like the figures on the contribution of the vGAT-GCaMP experiments, and have plateaued at the point where I think they are useful and interesting. But I think they are superficially described and illustrated, and conclude that this section should be strengthened. The vGluT2 data are so well illustrated. For example, Fig 1e (**most be 3e**) is really nice: the section/figure on inhibitory neurons would be well served by a figure like this.

We have added a Vgat version of Fig. 3e to Supplementary Figure 10 to meet the request of the reviewer.

7. Locomotion: given that the cord was cut at L3/4, it's hard to define the rhythmic activity as “locomotion” per se, in which you need alternating activity in flexor and extensor motor neurons. That is, while I think it's okay to use the term most of the time as the evidence is good, I think this caveat should be expressed at the outset.

We do not think we call it ‘locomotion’ here. We agree that we only measure flexor related activity. We know from 25 years of experiments with the *in vitro* preparation that when flexor related roots are active, they are out of phase with extensors which we normally call locomotor like activity because it corresponds to a complex flexor extensor pattern in the hindlimb (Kiehn and Kjaerulff 1996).

8. Standard error of the mean: This is a completely useless number and should be eliminated from all figures. In biology, we're interested in variability, not where a “true” mean is. As such, at minimum, the standard deviation should be shown. Better would be box-whisker plots with medians, interquartile ranges, etc.

All the bar graphs are replaced with box-whisker plots according to reviewers' recommendation (Fig. 1o, Fig. 2o, Fig. 3f, Fig. 5e-f, Supplementary Fig. 6g, Supplementary Fig. 8i, and Supplementary Fig. 10c).

9. Modules: I think it's okay to use this word. I'll just point out that I think to many (or

at least to me), the word refers to particular microcircuits (see, e.g., El Manira zebrafish work), which weren't studied here (i.e. not at the neuronal level). So while I would go for the word "regions" here, it's up to the authors.

We opt to keep the word modules but are certainly aware of the general discussion.

Minor:

One general, minor comment that does not need a response. The 5HT work was very interesting. While I think most investigators would say that the descending command is glutamatergic (as the authors cite), many would say that 5HT plays an important auxiliary role. As such, Suppl fig 3 is very interesting in that there does not seem to be a modulating effect. It would be interesting, one day, to stimulate LPGi and add 5HT blockers!

Thanks for this comment – we agree and will keep it in mind.

1. Title: Grammatically incorrect, and should read "Deconstructing" for example.

Thanks for pointing it out. Title is now changed.

2. Introduction, 1st paragraph. The term "executive" is incorrect. When it comes to the CNS, executive function involves higher cognitive thought, and I don't think the authors mean that's what the spinal cord is doing. I think they mean, perhaps, "executing." Also in line 154. Texts are changed according to reviewer's comment at multiple places in texts.

3. Line 42, remove "that"

Texts are changed according to reviewer's comment (Page 2, Line 42).

4. Line 102 – locomotor efficiency, and locomotor index in the figure – I cannot find how efficiency was defined, or how index was calculated.

The definition of locomotor index was presented in the lower panel of Supplementary Figure 1b. We have now moved the panel to the middle of the figure. In terms of efficiency, we show that rostral 12N contains more locomotor-initiating spots (indicated by red circles) than other rostrocaudal levels in both Supp. Fig. 2 and 3, which is an indication that rostral 12N is the most effective site.

5. Line 179, perhaps defined instead of confined?

Texts are changed according to reviewer's comment (Page 8, Line 195).

6. Figure 2n is not clear, and the legend doesn't help much. Could be improved.

Explanation is now expanded in the legend.

7. Perhaps it's simply my lack of expertise, but the authors are suggesting repeatedly that calcium signal can be seen in descending axons. How does this work? How is the calcium entering? I suppose that these RSN axons have calcium channels (what kind,

for what?) at their nodes of Ranvier, or are perhaps still unmyelinated at this age? It might be nice to add a couple of statements about this somewhere. Has a similar finding of Ca activity in long axons been previously described?

We thank the reviewer for this very thoughtful comment. It is not readily understood. It has been shown that there are some calcium channels in the nodes of Ranvier axons, but we tend to think that because the axons are still unmyelinated they might have a distributed complement of calcium channels throughout the axons. We now refer to relevant papers in the discussion (Page 19, Line 487-490). In all cases, it is very clear that the descending axons light up strongly in the funiculus and in the brainstem (we actually see the bundle of axons light up in the lower brainstem as they descend to the spinal cord (Supplementary. Fig. 1e) when the cell bodies are activated.

8. Line 220, quantify “slower onset”

We now have quantified the slower onset (Page 10, Line 239).

9. Line 259: when reading this, one wonders why it takes 100 ms. Much later, it becomes clear that that’s the time resolution of the experiments. Best to mention that here so the reader can interpret.

We have changed the texts to fit the reviewer’s comment (Page 11, Line 276-279).

10. Lines 349-350 – just a comment about some preps vs others – this could reflect the clustering of these neurons in the rostro-caudal dimension, and sometimes you don’t hit the clusters?

It could be, but it could also just be differences between preparations.

11. Line 353 – not clear how the authors can say definitively that these are “non locomotor initiating”.

We agree with the reviewer and now have deleted the text (Page 16, Line 386).

12. Lines 389-390: “without having to penetrate tissue” is a false argument. Penetration does little harm, and certainly much less than lopping off rostral regions! We did not mean that, but rather that the electrode was placed under visual control. The texts are now deleted to avoid confusion (Page 18, Line 418).

13. Line 395: “impossible” is a pretty strong word

We have changed the texts to fit the reviewer’s comment (Page 18, Line 424).

14. Line 438, maybe concurrently instead of coherently?

Texts are changed according to reviewer’s comment (Page 19, Line 481).

Methods:

1. Dissection – temperature of recordings should be indicated

Now given (Page 23, Line 630)

2. Lines 585-586, verifying level of the cut – this isn't clear at all. How?

We now clarified this issue on Page 23, Line 639-641.

3. The nonparametric tests were not defined. When and how often were the data normally distributed?

The data are normally distributed. It was mistake to mention nonparametric tests because they are not used in the current version (Page 26, Line 767-768).

REVIEWERS' COMMENTS

Reviewer #1 (Remarks to the Author):

I have read the revision of the paper by Hsu et al and I am satisfied with their changes and comments.

Regarding the L5 imaging experiment, I tend to agree with the authors (and with reviewer 3). It would be a good thing to know, and I suggested it in the hope that it had been done already and was lying dormant in some drawers, but in the economy of this paper it is not strictly needed.

On the other hand, the experiment with ventral root stimulation on the VGlut2-GCaMP mice is really important and I am glad it has been done to a high standard, with the inclusion of the control with ChAT-GCaMP mice. The results are neat and clarify the whole paper (and apologies for having to redact my initial comment).

Overall, I am very happy with the revisions and I do not have any further comment

Reviewer #2 (Remarks to the Author):

The authors responded fully to the critiques and I fully support acceptance of this manuscript for publication.

Reviewer #3 (Remarks to the Author):

The authors have done an excellent job addressing my comments on the previously submitted version. This manuscript significantly adds to our understanding of a key interaction between brain stem and spinal cord circuits. I have no further comments.